# Giant thermoelectric power factor in ultrathin FeSe superconductor

Sunao Shimizu[1], Junichi Shiogai[2], Nayuta Takemori[1], Shiro Sakai[1], Hiroaki Ikeda[3], Ryotaro Arita[1], Tsutomu Nojima[2], Atsushi Tsukazaki [2] & Yoshihiro Iwasa[1,4]

The thermoelectric effect is attracting a renewed interest as a concept for energy harvesting technologies. Nanomaterials have been considered a key to realize efficient thermoelectric conversions owing to the low dimensional charge and phonon transports. In this regard, recently emerging two-dimensional materials could be promising candidates with novel thermoelectric functionalities. Here we report that FeSe ultrathin films, a high-$T_c$ superconductor ($T_c$; superconducting transition temperature), exhibit superior thermoelectric responses. With decreasing thickness $d$, the electrical conductivity increases accompanying the emergence of high-$T_c$ superconductivity; unexpectedly, the Seebeck coefficient $\alpha$ shows a concomitant increase as a result of the appearance of two-dimensional natures. When $d$ is reduced down to ~1 nm, the thermoelectric power factor at 50 K and room temperature reach unprecedented values as high as 13,000 and 260 $\mu$W cm$^{-1}$ K$^{-2}$, respectively. The large thermoelectric effect in high $T_c$ superconductors indicates the high potential of two-dimensional layered materials towards multi-functionalization.

[1] RIKEN Center for Emergent Matter Science (CEMS), Wako, Saitama 351-0198, Japan. [2] Institute for Materials Research, Tohoku University, Sendai 980-8577, Japan. [3] Department of Physics, Ritsumeikan University, Kusatsu, Shiga 525-8577, Japan. [4] Quantum Phase Electronics Center (QPEC) and Department of Applied Physics, University of Tokyo, Bunkyo, Tokyo 113-8656, Japan. These authors contributed equally: Sunao Shimizu, Junichi Shiogai. Correspondence and requests for materials should be addressed to Y.I. (email: iwasa@ap.t.u-tokyo.co.jp)

Two-dimensional (2D) materials are expanding their arena in terms of richness in material type, properties, and functions, which range from electronic devices to catalysts and medicines[1,2]. Thermoelectric generation is one of the physical functions in which 2D materials are anticipated to be superior in comparison with their bulk counterparts. The density of states (DOS) in 2D semiconductors is considerably different from that of three-dimensional (3D) materials at the band edge singularity[3]. As the Seebeck coefficient $\alpha$ is related to the profile of the DOS at the Fermi energy, 2D or low dimensional structures are considered to be advantageous for enhancing thermoelectric performance. Such a concept was proposed originally for semiconductor quantum wells and superlattices;[4] however, recently emerging 2D-layered materials provide naturally formed atomic layers and their hetero-structures[5], which are an ideal platform to elicit their intrinsic 2D nature. For characterization of thermoelectric properties of nanomaterials, on-chip device measurements have been often utilized[6–8]. Although the device configuration used for the measurements is not directly adapted to practical applications, it is highly powerful for realizing ideal conditions including the structures free from significant disorder and the tunable carrier density and thus for elucidating the intrinsic performance of materials. This method also fits the thermoelectric characterization of 2D materials in the present study.

The performance of thermoelectric semiconductors is measured by the figure of merit $ZT = \alpha^2 T/\rho\kappa$ (where $\rho$ is the electrical resistivity, $\kappa$ is the thermal conductivity, and $T$ is the absolute temperature). Therefore, materials with the large power factor $\alpha^2/\rho$ can be candidates for high $ZT$. In order to maximize $\alpha^2/\rho$, we propose to extensively investigate recent 2D layered materials. In addition to the possible enhancement of the Seebeck effect in 2D DOS, an important characteristic of the recent 2D materials is their excellent crystallinity, which is preferable for keeping a large conductivity even in nano-thick monolayers.

For our purpose, 3d transition-metal-based compounds should be more favorable than 4d and 5d counterparts because the wave functions of 3d-based compounds are more localized, generally causing a larger effective mass $m^*$ and thus the larger DOS. Among 3d-based materials, we chose FeSe, first because a relatively large $m^*$ ranging from 2 to 4 $m_e$ has been reported in heavily electron-doped regions[9,10], where $m_e$ is the free electron mass. The physical properties of ultrathin FeSe have attracted much attention because of the appearance of the unexpected high-$T_c$ superconducting phase by reducing the film thickness down to a monolayer, the $T_c$ of which reaches 65 K[11,12] or 100 K[13]. Surprisingly, the high conductivity value survives even in monolayer FeSe;[11,14,15] this is in stark contrast to conventional semiconductor thin films, where the resistance increases with reducing the thickness.

Here we report simultaneous measurements of $\alpha$ and $\rho$ while controlling the thickness $d$ of FeSe films on SrTiO$_3$ (001) substrates in an electric double-layer transistor configuration[16]. In previous studies, we succeeded in optimization of $\alpha^2/\rho$ with controlling $n$ through the gate bias $V_G$ and applied this technique to various materials[16] (see Methods). When $V_G$ is applied at ~220 K, which is just above the glass transition temperature of the ionic liquid used in this study (see Methods), the cations or anions are self-aligned on the surface of FeSe; thus, charge carriers are electrostatically accumulated to form the electric double layer[17,18]. On the other hand, when a certain level of $V_G$ is applied at higher temperatures such as ~245 K or above, an electrochemical reaction takes place at the liquid–solid interface, and the topmost FeSe layer dissolves into the ionic liquid in a pseudo layer-by-layer manner[15]. Therefore, systematic investigation of the thermoelectric properties from bulk to ultrathin FeSe now becomes possible at a wide temperature range from 10 K to around room temperature. We found that the thermoelectric effect is dramatically enhanced with reducing $d$ down to ~1 nm and thermoelectric power factor at 50 K and room temperature reach unprecedented values as high as 13,000 and 260 $\mu$W cm$^{-1}$ K$^{-2}$, respectively. The coexistence of giant thermoelectric power factor and high-$T_c$ superconductivity indicates the high potential of 2D layered materials towards multi-functionalization.

## Results

**Electrochemically enhanced Seebeck effect in FeSe thin film.** Dimensionality is a possible key factor to induce the evolution of the thermoelectric response owing to the characteristic DOS (Fig. 1a, b). The electric double layer transistor configuration shown in Fig. 1c enables us to control the film thickness $d$ through the electrochemical etching on the surface of the FeSe films (Fig. 1d). Figure 2a shows the thermoelectric voltage $\Delta V$ as a

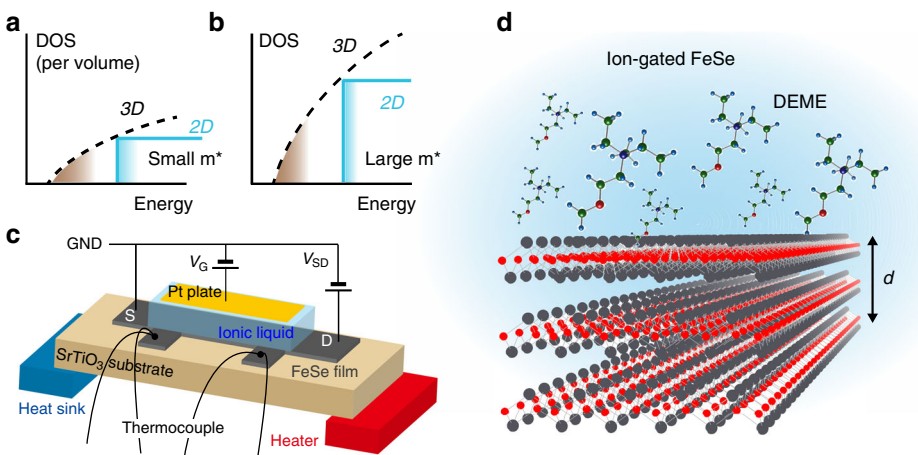

**Fig. 1** Schematic device structure for thermoelectric measurement. **a** Schematic illustration of the electronic density of states (DOS) for three-dimensional (3D) and two-dimensional (2D) electrons. **b** The large effective mass $m^*$ enhances the DOS, which is favorable to the enhancement of the Seebeck effect. **c** Device structure for thermoelectric measurement. $V_{SD}$ and $V_G$ stand for the source (S)−drain (D) voltage and the gate bias voltage, respectively. When $V_G$ is applied to the Pt plate, ions in the ionic liquid are redistributed, forming an electric double layer on the surface of the FeSe film. **d** Enlarged illustration of the ionic liquid/FeSe interface. Under the positive gate bias, $N,N$-diethyl-$N$-(2-methoxyethyl)-$N$-methylammonium cations, DEME +, align on the surface of FeSe. The thickness $d$ of the FeSe thin film was tuned by electrochemical etching[15]. See Methods for details of the device structure and fabrication

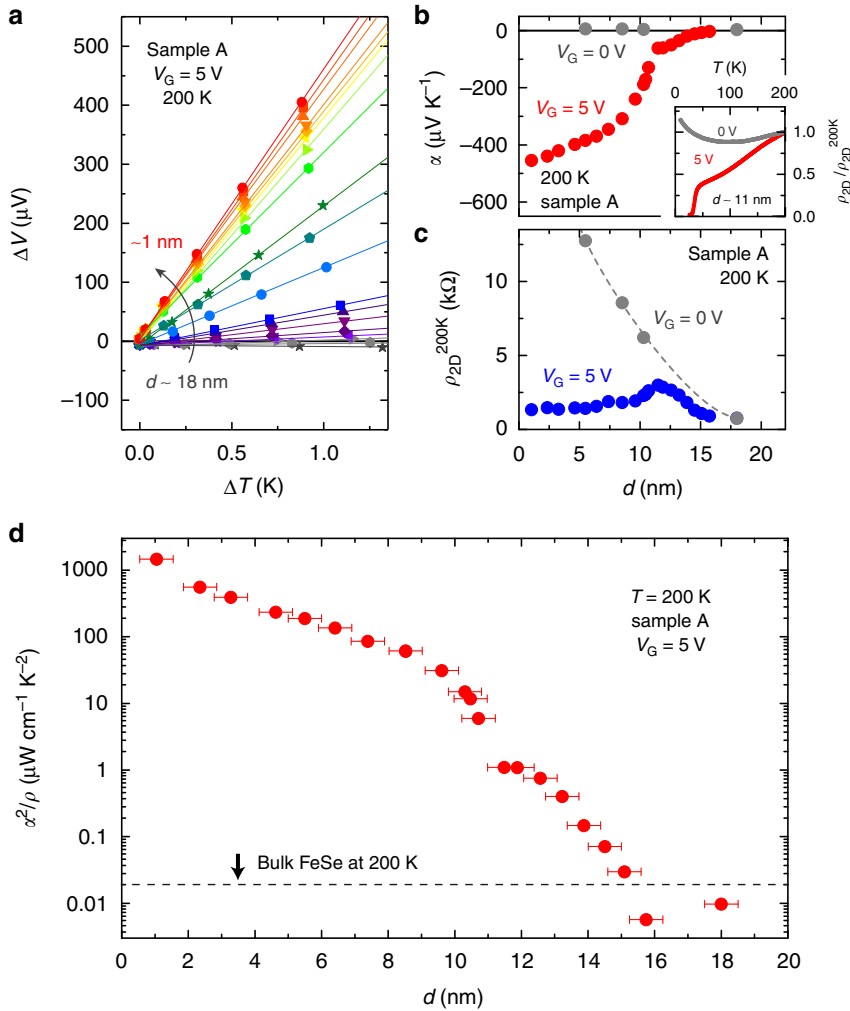

**Fig. 2** Thinning-induced enhancement of thermoelectric effect. **a** Thermoelectric voltage $\Delta V$ under temperature difference $\Delta T$ in an FeSe thin film, Sample A. The thermoelectric measurements were performed at 200 K with $V_G = 5$ V. The value of $\Delta V$ changed its sign and dramatically increased with decreasing thickness $d$ from ~18 nm to ~1 nm. **b** Thickness $d$ dependence of the Seebeck coefficient $\alpha$. The values of $\alpha$ were estimated from the slope of the $\Delta V - \Delta T$ plot in **a** as $\alpha = -\Delta V / \Delta T$. The inset shows the temperature $T$ dependence of the normalized sheet resistance $\rho_{2D}$ with respect to 200 K. Superconductivity appeared when $V_G = 5$ V was applied. **c** Variation of sheet resistance $\rho_{2D}{}^{200K}$ at 200 K as a function of $d$. The value of $\rho_{2D}{}^{200K}$ for $V_G = 5$ V showed a weak $d$ dependence (blue circles), whereas that for $V_G = 0$ V increased with decreasing $d$ (gray circles). **d** Thickness $d$ dependence of thermoelectric power factor $\alpha^2/\rho$ exhibiting anomalous enhancement in the ultrathin limit. Here, $\rho$ is the electrical resistivity, which is estimated as $\rho = \rho_{2D} \times d$. The value of $\alpha^2/\rho$ at 200 K in the thick region is comparable to that in bulk FeSe[19], whereas $\alpha^2/\rho$ increased with decreasing $d$ because of the double-digit increase of $\alpha$ in ultrathin regions in **b**. The error bar of $d$ corresponds to ~ ± 0.5 nm, which was estimated from the surface roughness of the initial thin film[15]

function of the temperature difference $\Delta T$ between two thermocouples (see Fig. 1c) in Sample A at $V_G = 5$ V. Here, it is noted that the application of $V_G$ not only induces the electrochemical etching of the thin films but also accumulates the electron carriers on the top surface. The thermoelectric measurement was done at 200 K; possible conduction paths through the ionic liquid are completely eliminated. The device was cooled down to 200 K for the measurement at each thickness $d$ after electrochemical etching at higher temperatures[15]. All the $\Delta V$ plots for different $d$'s linearly depended on $\Delta T$, securing the accurate characterization of the Seebeck effect. At 200 K, $\Delta V$ was dramatically enhanced across the sign change with decreasing $d$ from ~18 nm to ~1 nm. Figure 2b plots the $d$ dependence of $\alpha$ ($= -\Delta V/\Delta T$) at 200 K for $V_G = 5$ V. A very small $\alpha$ of $+3.8$ μV K$^{-1}$ at $d$ ~18 nm is consistent with the reported values on bulk FeSe[19,20], where such a small $\alpha$ value reflects the semimetallic electronic structure[21]. With the thickness reduction, on the other hand, the absolute value of $\alpha$

was surprisingly enhanced by two orders of magnitude up to $|-454 \text{ μV K}^{-1}|$.

It should be noted that the parasitic conduction of the SrTiO$_3$ substrate is ruled out because the gate electric field on SrTiO$_3$ through the FeSe thin films is negligible in the present configuration owing to the screening effect in the metallic conducting FeSe and also the formation of the Schottky barrier at the FeSe/SrTiO$_3$ interface[22–24] (see Supplementary Figure 1, Supplementary Figure 2, and Supplementary Note 1). The existence of an oxygen deficient layer at the surface of SrTiO$_3$ substrate as a source of the large Seebeck response is also definitely ruled out as the large $\alpha$ is observed only under gate bias and is suppressed to bulk-like small values by switching off $V_G$ to 0 V, as seen in the main panel of Fig. 2b. Importantly, the high-$T_c$ superconductivity appears by applying $V_G = 5$ V and disappears by removing $V_G$[15,17,18], as shown in the inset of Fig. 2b. The simultaneous emergence of the giant thermoelectric response and

the high-$T_c$ superconductivity proves that these two transport properties arise from the same electronic state of FeSe thin films.

Another noticeable feature of FeSe thin films is the low electrical resistance realized even in ultrathin regions. Figure 2c shows the 2D sheet resistance $\rho_{2D}$ of Sample A for $V_G = 0$ V (gray circles) and 5 V (blue circles) as a function of $d$. When starting from the initial state with $d \sim 18$ nm, the sheet resistance at 200 K, $\rho_{2D}^{200 K}$, first increased with decreasing $d$ for both $V_G = 0$ V and 5 V. With further decreasing $d$, $\rho_{2D}^{200 K}$ at $V_G = 5$ V showed a small peak at around $d \sim 11$ nm and kept small values down to $d \sim 1$ nm because the gated topmost layer of FeSe and the charge transfer layer at the FeSe/SrTiO$_3$ interface dominate the electrical transport of the thin film (see Supplementary Figure 3 and Supplementary Note 2 for the details of the $d$ dependence of $\rho_{2D}^{200 K}$). Such a low electrical resistance irrespective of the film thickness is consistent with the previous studies; for example, the resistivity of monolayer or few layer MBE-grown FeSe[11,14] is comparable to that of 10 nm thick ($\sim$15 layers) FeSe[17,18] owing to the interface or surface electron doping. Actually, $\rho_{2D}^{200 K} \sim 1$ kΩ at $V_G = 5$ V in the thin limit (Fig. 2c) is close to that in doped FeSe monolayers[11,14,15]. On the other hand, the small $\alpha$ and high $\rho_{2D}^{200 K}$ at $V_G = 0$ V indicate that the charge transfer layer does not produce the enhanced values of $\alpha$. Consequently, the thermoelectric power factor $\alpha^2/\rho$ at 200 K achieved a dramatic development in Fig. 2d owing to the enhancement of $\alpha$ and the concomitant reduction of electrical resistivity $\rho = \rho_{2D} \times d$, which rarely occurs in the framework of

conventional material design and fabrication. Along with the reduction of $d$ from 18 nm to 1 nm, $\alpha^2/\rho$ kept increasing and finally reached $\sim$1500 μW cm$^{-1}$K$^{-2}$.

**Temperature-thickness mapping of thermoelectric response.** Figures 3a, b display the temperature $T$—thickness $d$ mappings of the absolute value of $\alpha$ (i.e., $|\alpha|$) and $\alpha^2/\rho$, respectively, for another FeSe thin film, Sample B. The values of $|\alpha|$ and $\alpha^2/\rho$ showed dramatic developments in the nanometer-thick region, which agrees well with the results for Sample A (see Figs. 2b, c). Moreover, the enhancement for both $|\alpha|$ and $\alpha^2/\rho$ covers a wide temperature range from 50 K (just above $T_c$) to 280 K. Figure 3c summarizes $\alpha^2/\rho$ for representative thermoelectric materials that possess high $\alpha^2/\rho$ values (see Supplementary Table 1). The values of $\alpha^2/\rho$ for the FeSe ultrathin film increased from $\sim$260 μW cm$^{-1}$ K$^{-2}$ at 280 K up to $\sim$13,000 μW cm$^{-1}$ K$^{-2}$ at 50 K, being the largest among existing bulk materials reported so far. Assuming the thermal conductivity for bulk Fe-based superconductors[25,26], $\kappa \sim 5$ W m$^{-1}$ K$^{-1}$, the dimensionless figure of merit $ZT$ of the FeSe ultrathin film reaches as large as $\sim$1.5 at 280 K.

**Common trend of Seebeck effect in Fe-based superconductors.** The detailed temperature dependence of $\alpha$ for different $d$'s is presented in Fig. 4a to show the unusual thermoelectric response in FeSe. Except for the initial thickness (19.1 nm) with moderate

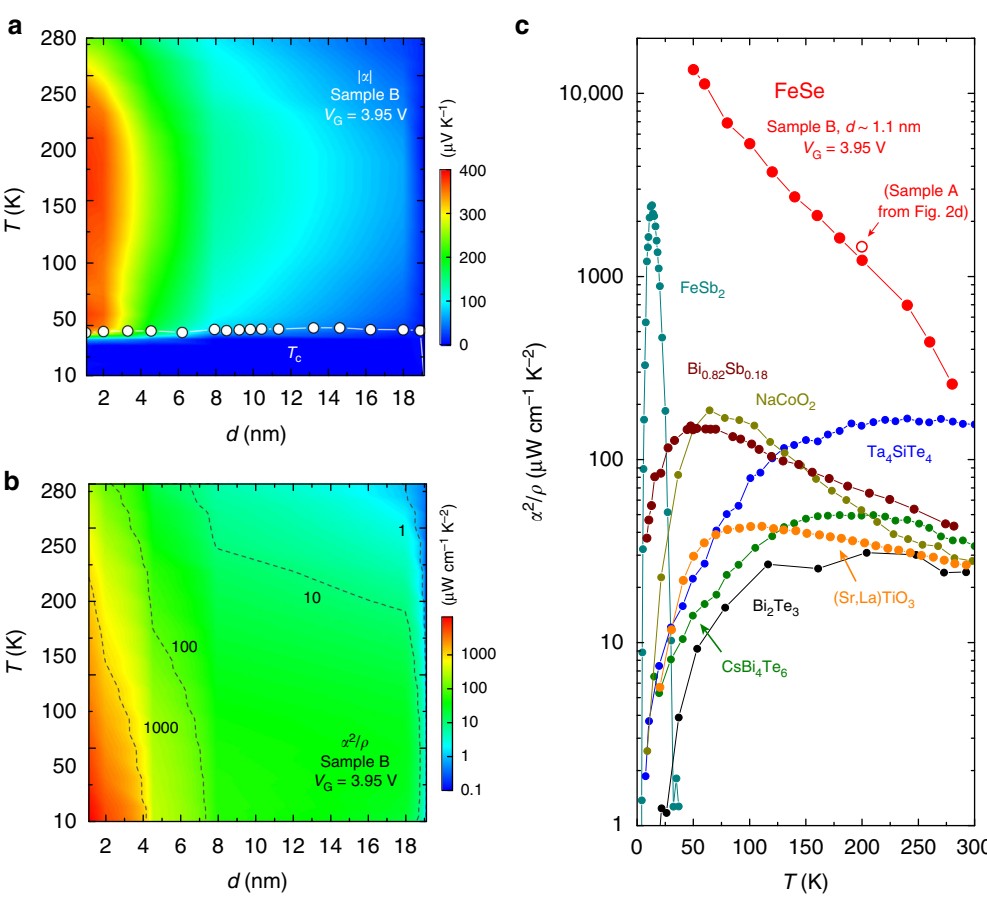

**Fig. 3** Giant thermoelectric response in ultrathin FeSe. **a** Mapping of Seebeck coefficient $\alpha$ of FeSe (Sample B) against temperature $T$ and thickness $d$. Here, the absolute value of $\alpha$ (i.g., $|\alpha|$) for $V_G = 3.95$ V was plotted. The white circles correspond to the onset temperature $T_c$ of the superconducting transition. **b** Evolution of thermoelectric power factor $\alpha^2/\rho$ of FeSe (Sample B) above $T_c$. The value of $\alpha^2/\rho$ increased with decreasing $d$, mainly owing to the large enhancement of $\alpha$ shown in **a**. **c** Comparison of temperature dependence of $\alpha^2/\rho$ among representative thermoelectric materials. The values of $\alpha^2/\rho$ in the FeSe ultrathin film were larger than any existing bulk materials reported so far in a wide temperature range (see Supplementary Table 1 for the reference of the experimental values). The data point for Sample A at 200 K (open circle) shows fair reproducibility

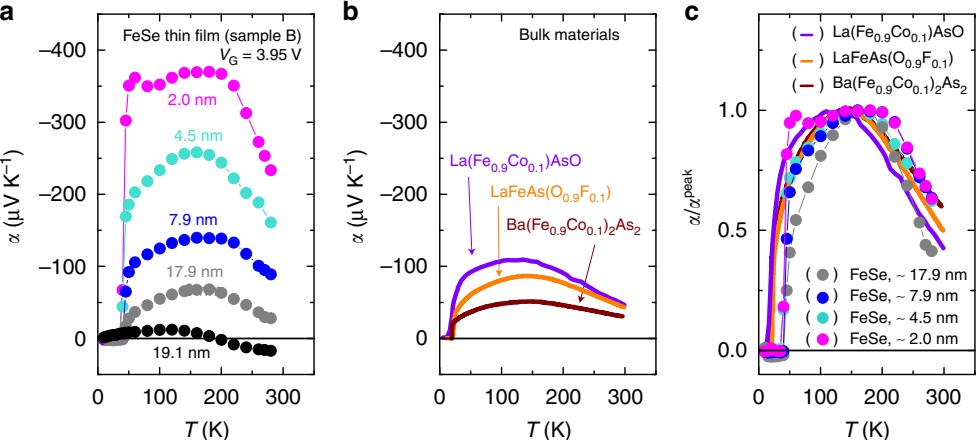

**Fig. 4** Temperature dependence of Seebeck coefficient. **a** Seebeck coefficient $\alpha$ vs. temperature $T$ in FeSe thin film (Sample B). The values of $\alpha$ were enhanced by the thickness reduction in all temperature regions. **b** $\alpha$–$T$ curves in representative Fe-based high-$T_c$ superconductors. Ba(Fe$_{0.9}$Co$_{0.1}$)$_2$As$_2$[27], LaFeAs(O$_{0.9}$F$_{0.1}$)[28], and La(Fe$_{0.9}$Co$_{0.1}$)AsO[29] show the peak behavior at ~150 K, in a similar manner to the FeSe thin film in **a**. **c** Temperature dependence of $\alpha$ normalized by the peak value $\alpha^{peak}$ for FeSe thin films and bulk materials in **b**. Overall temperature variation of $\alpha$ is seemingly common for Fe-based superconductors

temperature dependence, $\alpha$ for Sample B showed a peak at around ~200 K, which follows neither the $T$-linear behavior expected in conventional metals nor the phonon drag thermopower (see Supplementary Figure 1 and Supplementary Note 1). Actually, the temperature dependence of $\alpha$ in the FeSe thin film is qualitatively similar to that in bulk Fe-based high-$T_c$ superconductors such as Ba(Fe$_{0.9}$Co$_{0.1}$)$_2$As$_2$[27], LaFeAs(O$_{0.9}$F$_{0.1}$)[28], and La(Fe$_{0.9}$Co$_{0.1}$)AsO[29], as shown in Fig. 4b. This trend can be seen even more clearly in Fig. 4c, where the data in Fig. 4a, b are normalized by the peak value $\alpha^{peak}$ of each sample. These similarities further prove that $\alpha$ observed in Fig. 4a is attributed to FeSe itself rather than other artifacts such as substrates and ionic liquids. The characteristic temperature dependence of $\alpha/\alpha^{peak}$ in Fig. 4c is considered unique to Fe-based superconductors and has been discussed in the context of quantum criticality[30–32] or the two carrier model[27,33]. For example, it was reported that $|\alpha|/T$ in Ba(Fe$_{0.9}$Co$_{0.1}$)$_2$As$_2$[27] shows a divergence above $T_c$ and a strong enhancement when in proximity to the quantum critical point.

**Electronic band structure in gated FeSe thin film.** The transfer characteristics (the $V_G$ dependence of resistance) of a FeSe thin film (see Supplementary Figure 4 and Supplementary Note 3) indicates that the dominant carriers change from holes to electrons with reducing the thickness. This behavior is consistent with the band structure evolution derived from the angle-resolved photoemission spectroscopy (ARPES)[12,34–36] in monolayer FeSe on SrTiO$_3$ and in K-coated FeSe thin films, which clarified that the hole pocket at the $\Gamma$-point disappears and a gap of ~60 meV is opened at the M-point[34] owing to the thinning and concomitant electron doping. The present ion-gated FeSe thin films should have a similar band structure because the electron density accumulated by the ionic gating, ~$10^{14}$ cm$^{-2}$, is comparable to that of the charge transfer from SrTiO$_3$ substrate and of surface K coating. This band structure of FeSe monolayer should be beneficial for the enhancement of $|\alpha|$. We calculated the Seebeck coefficient for the undoped bulk FeSe and electron-doped monolayer FeSe at $T = 280$ K (see Supplementary Figure 5 and Supplementary Note 4), and obtained $\alpha$ values as $+5$ μV K$^{-1}$ and $-200$ μV K$^{-1}$ for the bulk and the monolayer FeSe, respectively. These estimations reasonably explain the experimental values of $\alpha$ at 280 K: $+17$ μV K$^{-1}$ and $-245$ μV K$^{-1}$ (Fig. 3a) for the initial ($d$ ~19.1 nm) and final ($d$ ~1 nm) thicknesses, respectively. On the other hand, our calculation based on the Fermi liquid picture

predicts $T$-linear behavior and does not explain the non-monotonous temperature dependence of $\alpha$ in ultrathin FeSe. The experimentally observed broad peak in $\alpha$ located at ~200 K (Fig. 4a) is suggestive of a crucial role of electronic correlations in the Seebeck response of ultrathin FeSe; in fact, the recent ARPES studies pointed out a strong electronic correlation[9] in the high-$T_c$ phase of FeSe. A quantitative theoretical analysis of this effect remains to be performed.

## Discussion

Nanostructures or low-dimensional structures have been a powerful guideline for the exploration of high-performance thermoelectric materials[8,37–40]. The present results show that further enhancement of thermoelectric properties should be possible, if peculiar band structures of nano-structured systems including 2D layered materials are combined with additional ingredients such as strong electronic correlations. The unprecedented coexistence of giant thermoelectric power factor and high-$T_c$ superconductivity in ultrathin FeSe exemplifies that there may exist unknown multifunctional materials waiting to be disclosed in extreme conditions, illuminating a next research direction of functional thermoelectric materials.

## Methods

**Device fabrication.** We fabricated ion-gated devices based on FeSe-thin films on SrTiO$_3$ substrates[15] with channel size of $1.2 \times 2$ mm$^2$. The details of the thin-film preparation were reported in our previous study[15]. The device structure used in this study is schematically shown in Fig. 1c. The FeSe thin films were patterned by using a laser cutter to perform four-terminal resistance measurements. The gold wires were attached at both edges of the patterned film, working as a drain terminal $D$ and a source terminal $S$. An ionic liquid, which worked as a gate dielectric, was placed on the FeSe surface. We used $N,N$-diethyl-$N$-(2-methoxyethyl)-$N$-methylammonium bis-(trifluoromethylsulfonyl)-imide (DEME-TFSI) as the ionic liquid. A Pt plate was placed on top of them, working as a gate electrode.

**Thermoelectric measurements under gate biases.** As shown in Fig. 1c, a heater and a heat sink were attached to either side of the ion-gated device to produce a thermal gradient. The type E thermocouples were attached to monitor the temperature difference $\Delta T$ and the thermoelectric voltage $\Delta V$. The thermocouples were also used for the four-terminal resistance measurements. The temperature difference $\Delta T$ (0–1 K) and the voltage $\Delta V$ between the thermocouples were measured, and the values of $\alpha$ were evaluated from the slope of the $\Delta V-\Delta T$ plots (See Fig. 2a). This device configuration allows us to measure $\alpha$ and $\rho$ simultaneously. The thermoelectric measurements with solid[41–47] and ionic gate dielectrics[48–57] are widely accepted as a method to evaluate the thermoelectric properties of semiconductors with changing the carrier densities.

**Calculations.** We performed first-principles band structure calculations using the Perdew-Burke-Ernzerhof parameterization of the generalized gradient approximation[58] and the full-potential (linearized) augmented plane-wave method, with the inclusion of spin-orbit coupling as implemented in the wien2k code[59]. Muffin-tin radii ($R_{MT}$) of 2.38 and 2.11 Bohr were used for Fe and Se, respectively. The maximum modulus for the reciprocal vectors $K_{max}$ was chosen such that $R_{MT}K_{max}$ = 7.0 and a $10 \times 10 \times 10$ k-mesh in the first Brillouin zone was used. The tight-binding Hamiltonian for the $3d$ orbitals of the Fe atom was constructed with Wannier90[60] and wien2wannier[61].

## Data availability

The authors declare that all data supporting the findings of this study are available within the paper and its Supplementary Information or from the authors upon reasonable request.

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

## Acknowledgements

This work was supported by JSPS KAKENHI Grant Numbers JP25000003, JP16H00923 (SATL), JP16H06345, JP17H02928, JP17K19060.

## Author contributions

S.Sh. and J.S. equally contributed to this work. S.Sh., J.S., T.N., A.T. and Y.I. conceived and designed this work. S.Sh., N.T., S.Sa., A.T. and Y.I. wrote the paper. S.Sh. and J.S. performed all the measurements, and N.T., S.Sa., H.I. and R.A. conducted the calculations. All authors contributed to discussions.

## Additional information

**Competing interests:** The authors declare no competing interests.

