## [Peer Review File · Nature Communications]

Reviewers' comments:

Reviewer #1 (Remarks to the Author):

This MS reports a giant powder factor of $13000 \mu\text{Wcm}^{-1}\text{K}^{-2}$ for ultrathin FeSe film with thickness of $\sim 1\text{nm}$. However, I have a strong objection in the conclusion, which attributes the high power factor to the ultrathin 2D FeSe film. Actually, the two parameters used to estimate the power factor, Seebeck coefficients and electrical resistivities (conductivities), are very different when measured with/without gate bias (Figure 2b,c, Page 6 line 106). For example, the Seebeck coefficient measured without gate bias is similar to bulk sample ($\sim 0 \mu\text{VK}^{-1}$) and are largely lower than that measured with gate bias ($\sim 400 \mu\text{VK}^{-1}$) for sample with thickness of $\sim 5\text{nm}$. But the measurement without gate bias is the routine test method in thermoelectric community. Therefore, the high powder factor observed here mainly originated from other parts introduced by the measure method instead of materials (FeSe). There is no reasonable explanation for the big difference in these different methods in this MS. In addition, the authors didn't consider the effect of the charge transfer layer at the FeSe/SrTiO₃ interface (Page 7 line 121, Ref.15) on the Seebeck coefficient and electrical conductivity. Accordingly, the comparison of powder factor obtained here with other materials measured by common method is unacceptable and meaningless (Figure 3c). Thus, I don't recommend publication of this work in Nature Communications.

In addition, other issues are summarized below:

1. Page 4, line 66. 'large m^* ranging from 2 to 4' should be 'large m^* ranging from 2 to 4 m_e '
2. Page 7 line 116 and Figure 2c. For the electrical conductivity calculation, the geometric size of the film is need, the authors need to provide the wide sizes for each films. In addition, on page 8 line 131, the electrical conductivity $\sigma=1/(R \times d)$ is incorrect.
3. Page 7 line 118 and Figure 2c. Given no electrical resistivity data was provided for samples with thickness between 1-5 nm and 10-18 nm measured without gate bias, thus 'the sheet resistance at 200 K, $R_{s200\text{K}}$, first increased with decreasing d for 120 both $V_G = 0\text{V}$ and 5V , as expected' is unconvincing.

Reviewer #2 (Remarks to the Author):

I have read the manuscript entitled "Giant thermoelectric power factor in ultrathin FeSe superconductor", which is submitted to Nature Communications by Shimizu et al. In this manuscript, the authors reduce the FeSe layer thickness by the electrochemical etching technique and measure the thickness dependences of thermoelectric properties above the T_c . The main claim is that the 1-nm-thick FeSe showed huge thermoelectric power factor ($26 \text{mW/m}^2\text{K}^2$ at RT, $1300 \text{mW/m}^2\text{K}^2$ at 50 K) due to its 2D nature (evolution of the electronic structure). Since these PF values are several orders of magnitude greater than the reported values of other thermoelectric materials, the manuscript is possibly worth for the publication in Nature Communications. There are several concerns. I will be satisfied if the authors answer my questions and comments listed below.

(1) After reading the manuscript, first, I doubted the data analyses because there is a possibility that the SrTiO₃ substrate contributed to the observed Seebeck coefficient since SrTiO₃ became conductor if oxygen deficiency is formed. But the authors checked that SrTiO₃ substrate did not contribute the observed Seebeck coefficient due to very high contact resistance between the FeSe and SrTiO₃ (Supplementary).

(2) Another concern is the contribution of the FeSe/SrTiO₃ interface. Although the authors have checked the SrTiO₃ itself did not contribute to the observed Seebeck coefficient, the authors did not mention that there is an interface which could contribute to the observed Seebeck coefficient. Since it is known that 2DES at the heterointerface at an insulator/SrTiO₃ exhibits a rather large Seebeck coefficient (Refs.), the authors may need to consider the contribution of the

heterointerfaces FeSe/SrTiO₃.

Ohta, H. et al., *Nature Mater.* 6, 129 (2007).; Choi, W.S. et al., *Phys. Rev. B* 82, 024301 (2010).; Ohta, H. et al., *Nature Commun.* 1, 118 (2010).; Ohta, H. et al., *Adv. Mater.* 24, 740 (2012).; Zhang, Y. et al., *Nature Commun.* 9, 2224 (2018).

(3)The best way to rule out the possibility of SrTiO₃ contribution, the authors can simply use MgO or LaAlO₃ single crystal as the substrate because MgO and LaAlO₃ are insulators do not contribute to the Seebeck coefficient. I am wondering why the authors did not use such insulating substrates though the authors used MgO substrate in the previous study [Shiogai, J. et al., *Nature Phys.* 12, 42 (2016)].

(4)As the authors mentioned in the introduction part, low thermal conductivity is also required for the good thermoelectric materials. For the readers, it should be better if the authors comment on the thermal conductivity of FeSe.

(5)Some minor questions and comments

(5-1) In the abstract, the authors used S as the Seebeck coefficient whereas they used α in the main text.

(5-2) The size of the devices should be given in the Method section.

(5-3) The type of the thermocouple should be given in the Method section.

Response to Reviewers' Comments

First of all, we thank Reviewers for their careful reviewing of our manuscript. We have revised the manuscript according to the reviewers' comments. We would like to reply to all the comments given by the two reviewers in the followings. In addition, for the reviewing convenience, we attached the main text and Supplementary Information, in which all the revised points are highlighted in yellow, at the end of this response letter.

*** Reply to comments of Reviewer#1 ***

[Comment 1-1]

This MS reports a giant powder factor of 13000 $\mu\text{Wcm}^{-1}\text{K}^{-2}$ for ultrathin FeSe film with thickness of $\sim 1\text{nm}$. However, I have a strong objection in the conclusion, which attributes the high power factor to the ultrathin 2D FeSe film. Actually, the two parameters used to estimate the power factor, Seebeck coefficients and electrical resistivities (conductivities), are very different when measured with/without gate bias (Figure 2b,c, Page 6 line 106). For example, the Seebeck coefficient measured without gate bias is similar to bulk sample ($\sim 0 \mu\text{VK}^{-1}$) and are largely lower than that measured with gate bias ($\sim -400 \mu\text{VK}^{-1}$) for sample with thickness of $\sim 5\text{nm}$. But the measurement without gate bias is the routine test method in thermoelectric community. Therefore, the high powder factor observed here mainly originated from other parts introduced by the measure method instead of materials (FeSe). There is no reasonable explanation for the big difference in these different methods in this MS.

[Reply 1-1]

First of all, we would like to thank Reviewer#1 for the careful reviewing of our manuscript. As commented by the reviewer, the Seebeck effect of the FeSe thin film was measured with the field effect transistor configurations in this study. We are

Figure R1 Schematic pictures of conventional solid gate field effect transistor (a) and ion gated transistor (b). In both configurations, the positive gate bias electrostatically accumulates electron carriers on the surface of the semiconductors.

sorry for the lack of explanation for the device operation and measurement scheme, which may have confused the reviewer. Here we explain an n-type operation of a field effect transistor (FET). Figure R1a shows a schematic picture of a conventional solid gate FET. By applying a positive gate bias V_G , the electron charge carrier density at the surface of semiconductor channels is increased, resulting in the modulation of the electrical conductivity and the Seebeck effect for the gated topmost surface. Figure R1b shows an FET with an ionic liquid as a gate dielectric. Although the gate dielectric is changed from solid to liquid, the operation mechanism is basically the same. The difference is the amount of the accumulated charge carriers. In the case of liquid gate transistors, the accumulated carrier density is as large as $10^{14} \sim 10^{15} \text{ cm}^{-2}$, which is more than one order of magnitude larger than that of solid gate FETs, because electric double layers formed on the surface of semiconductor channels work as nano-gap capacitors.

We fabricated FeSe-based transistors shown in Fig. 1c and measured the two-dimensional sheet resistance and the Seebeck effect with and without gate bias. This type of measurement with transistor configurations is widely accepted as the

method to systematically evaluate the Seebeck coefficient of semiconductors with varying the carrier density. Actually, there are many experimental reports on the electric field effect control of the Seebeck effect using solid-gate dielectrics [N. Zavaritsky, *Physica B+C* **126**, 369 (1984); B. L. Gllagher *et al.*, *Semicond. Sci. Technol.* **2**, 456 (1987); J. P. Small *et al.*, *Phys. Rev. Lett.* **91**, 256801 (2003); K. P. Pernstich *et al.*, *Nature Mater.* **7**, 321 (2008); A. Yoshikawa *et al.*, *Appl. Phys. Express* **2**, 121103 (2009); H. Ohta *et al.*, *Appl. Phys. Lett.* **95**, 113505 (2009); Y. M. Zuev *et al.*, *Phys. Rev. Lett.* **102**, 096807 (2009)]. In addition, the ionic liquid gating has been also applied to various semiconductors: oxide semiconductors such as SrTiO₃ [S. Shimizu *et al.*, *Phys. Rev. B* **92**, 165304 (2015).], ZnO [S. Shimizu *et al.*, *PNAS* **113**, 6438 (2016).], and Cu₂O [R. Takayanagi *et al.*, *Appl. Phys. Express* **53**, 111101 (2014).], transition-metal chalcogenides [M. Yoshida *et al.*, *Nano Lett.* **16**, 2061 (2016).], graphene [Y. Chien *et al.*, *Scientific Reports* **6**, 20402 (2016).], and single-walled carbon nanotubes [K. Yanagi *et al.*, *Nano Lett.* **14**, 6437 (2014).]. To remove the confusion about the validity of measurements, in the revised manuscript, we have added a following sentence in the Method section: “The thermoelectric measurements with solid⁴²⁻⁴⁸ and ionic gate dielectrics⁴⁹⁻⁵⁸ is widely accepted as a method to evaluate the thermoelectric properties of semiconductors with changing the carrier densities.”

The reason why the Seebeck effect of the FeSe thin film is enhanced by applying gate bias V_G is that the electron carrier doping by V_G dramatically changes the electronic structure. This is nothing but what we are aiming at in this study.

This carrier doping effect was well discussed with the direct measurement of the variation of electronic states by angular resolved photoemission spectroscopy (APES) [D. Liu *et al.*, *Nature Commun.* **3**, 931 (2012).; S. He *et al.*, *Nature Mater.* **12**, 605 (2013).; J. He *et al.*, *PNAS* **111**, 18501, (2014).; Y. Miyata *et al.*, *Nature Mater.* **14**, 775 (2015).]. The electronic band structure of FeSe thin films is modified by electron doping from a semimetal to a gapped semiconductor, as schematically shown in Fig. R2. In the case of the semimetallic band structure (bulky, low T_c

Figure R2 Schematic band structure of FeSe. a, The electron carrier doping induces the reconstruction of the band structure from a semimetal (a) to a gapped semiconductor (b).

superconductor), holes at Γ point and electrons at M point contribute oppositely to the total Seebeck signal, which results in the small Seebeck effect. On the other hand, when the band gap is opened with heavily electron doped at M point (high T_c superconductor), the compensation of the Seebeck effect by holes and electrons is removed, resulting in the enhancement of the Seebeck effect. In addition, the carrier doping to FeSe enhances the effective mass [C.H.P. Wen *et al.*, Nature Commun. **7**, 10840 (2016).; J.J. Seo *et al.*, Nature Commun. **7**, 11116 (2016).], which is one of reasonable origins for the large enhancement of the Seebeck effect in our study.

We calculated the Seebeck coefficient of FeSe for both the semimetallic band structure and the gapped semiconducting one in Supplementary Information. We found that the Seebeck effect at $T=280$ K is enhanced from +5 to -200 $\mu\text{V/K}$ by the band reconstruction. This calculation reasonably explains the experimental values at $T=280$ K; + 17 $\mu\text{V/K}$ and -245 $\mu\text{V/K}$ for the initial and final thickness, respectively.

[Comment 1-2]

In addition, the authors didn't consider the effect of the charge transfer layer at the FeSe/SrTiO₃ interface (Page7 line 121, Ref.15) on the Seebeck coefficient and electrical conductivity. Accordingly, the comparison of powder factor obtained here with other materials measured by common method is unacceptable and meaningless (Figure 3c). Thus, I don't recommend publication of this work in Nature Communications.

[Reply 1-2]

In the manuscript, we considered the contribution of interface charge transfer layer of SrTiO₃ substrate in main text and supplement. We confirmed that the electron charge transfer from SrTiO₃ to FeSe does not induce a large Seebeck coefficient except for when the FeSe film becomes ultrathin. As for superconductivity in ultrathin FeSe, charge transferred electrons cannot penetrate into above 1 nm, and thus the emergence of high- T_c superconductivity is limited in ultrathin regime [Q. Wang et al., Chin. Phys. Lett. 29, 037402 (2012); S. Tan et al., Nature Mater. 12, 634 (2013); X. Liu et al., Nature Commun. 5, 5047 (2014)]. As shown in Figure 2, the large Seebeck coefficient was not observed when $V_G = 0$ V. If the FeSe/SrTiO₃ interface and the charge transfer layer contribute to the large Seebeck effect, the Seebeck coefficient should be large even at $V_G = 0$ V.

To make this point clearer, we added a following sentence in page 8, "On the other hand, the small α and high $\rho_{2D}^{200\text{ K}}$ at $V_G = 0$ V indicate that the charge transfer layer does not produce the enhanced values of α ."

[Comment 2]

Page 4, line 66. 'large m^* ranging from 2 to 4' should be 'large m^* ranging from 2 to 4 m_e '

[Reply 2]

We appreciate this comment. The description has been revised according to the reviewer's comment. We replaced "2 to 4" with "2 to 4 m_e ".

[Comment 3]

Page 7 line 116 and Figure 2c. For the electrical conductivity calculation, the geometric size of the film is need, the authors need to provide the wide sizes for each films. In addition, on page 8 line 131, the electrical conductivity $\sigma=1/(R \times d)$ is incorrect.

[Reply 3]

We are sorry for the confusing expression of σ . In the original manuscript, the two dimensional unit of R_s and three dimensional σ were used, namely the sheet resistance and conductivity, respectively. The expression for the sheet resistance is usually applied to two dimensional systems and transistor studies.

The calculation detail is following. The size of the channel area of our sample is $1.2 \times 2 \text{ mm}^2$. We have mentioned the channel size in Method section of the revised manuscript. In the previous version of the manuscript, the definition of R_s is $R_s = R_4 \times w / \ell$, where R_4 is the four terminal resistance, w is the width of the channel, and ℓ , is the length of the channel.

As pointed out by the reviewer, we now consider that we should have been more careful to avoid any ambiguities regarding the definition and notation of the physical properties. For this, in the revised manuscript, we use ρ_{2D} and ρ for the two dimensional sheet resistance and three dimensional resistivity, respectively. In addition, the expression for three dimensional conductivity σ is replaced by $1 / \rho$.

We hope that the paper becomes much clearer for readers.

[Comment 4]

Page 7 line 118 and Figure 2c. Given no electrical resistivity data was provided for samples with thickness between 1-5 nm and 10-18 nm measured without gate bias, thus ‘the sheet resistance at 200 K, $R_{s200\text{ K}}$, first increased with decreasing d for 120 both $V_G = 0\text{ V}$ and 5 V , as expected’ is unconvincing.

[Reply 4]

Thank you for this comment. Unfortunately, the data for $d = 18\text{ nm}$ without gate bias was overlapped with that with $V_G = 5\text{ V}$ in Figure 2c. We did not notice this point. In the revised manuscript, the data point for $d = 18\text{ nm}$ with $V_G = 0\text{ V}$ is plotted at front part to that of $V_G = 5\text{ V}$.

The expression of “as expected” is relevant on the sheet resistance ρ_{2D} shown in Fig. 2c, $\rho_{2D} = R_4 \times w / \ell = \rho / d$, where R_4 is the four terminal resistance, ρ is the three dimensional resistivity, d is the thickness. Therefore, it is natural that ρ_{2D} increases with decreasing d , if we assume homogeneous three dimensional three dimensional ρ dominates R_4 . Such behavior is a general tendency in thin films, indicating that the electrical conduction in the thicker region 10 – 18 nm is dominated by the bulky charge carrier due to weak gate effect. Actually, the two dimensional sheet resistance of FeSe thin films without gate bias was reported elsewhere [R. Schneider *et al.*, PRL **108**, 257003 (2012); J. Shiogai *et al.*, Nature Phys. **12**, 42 (2016).], which are consistent with our data.

In the revised manuscript, we have extended the dashed line up to $d = 18\text{ nm}$ in Fig. 2c and cited the above references.

*** Reply to comments of Reviewer#2 ***

[Comment 1]

I have read the manuscript entitled “Giant thermoelectric power factor in ultrathin FeSe superconductor”, which is submitted to Nature Communications by Shimizu et al. In this manuscript, the authors reduce the FeSe layer thickness by the electrochemical etching technique and measure the thickness dependences of thermoelectric properties above the T_c . The main claim is that the 1-nm-thick FeSe showed huge thermoelectric power factor (26 mW/m K² at RT, 1300 mW/m K² at 50 K) due to its 2D nature (evolution of the electronic structure). Since these PF values are several orders of magnitude greater than the reported values of other thermoelectric materials, the manuscript is possibly worth for the publication in Nature Communications. There are several concerns. I will be satisfied if the authors answer my questions and comments listed below.

[Reply 1]

We would like to thank Reviewer#2 for the careful reading of our manuscript. We are very happy that Reviewer#2 acknowledged the importance of our paper and considered it worth publication in Nature Communications. We would like to try our best to reply all the comments point by point.

[Comment 2]

After reading the manuscript, first, I doubted the data analyses because there is a possibility that the SrTiO₃ substrate contributed to the observed Seebeck coefficient since SrTiO₃ became conductor if oxygen deficiency is formed. But the authors checked that SrTiO₃ substrate did not contribute the observed Seebeck coefficient due to very high contact resistance between the FeSe and SrTiO₃ (Supplementary).

[Reply 2]

We appreciate this comment. As mentioned by Reviewer#2, our careful inspection for the contribution of SrTiO₃ substrate is reported in Supplementary Information. We are sure that the experiments are solid evidence for the negligible contribution of SrTiO₃ in the observed Seebeck coefficient.

[Comment 3]

Another concern is the contribution of the FeSe/SrTiO₃ interface. Although the authors have checked the SrTiO₃ itself did not contribute to the observed Seebeck coefficient, the authors did not mention that there is an interface which could contribute to the observed Seebeck coefficient. Since it is known that 2DES at the heterointerface at an insulator/SrTiO₃ exhibits a rather large Seebeck coefficient (Refs.), the authors may need to consider the contribution of the heterointerfaces FeSe/SrTiO₃.

Ohta, H. et al., Nature Mater. 6, 129 (2007).; Choi, W.S. et al., Phys. Rev. B 82, 024301 (2010).; Ohta, H. et al., Nature Commun. 1, 118 (2010).; Ohta, H. et al., Adv. Mater. 24, 740 (2012).; Zhang, Y. et al., Nature Commun. 9, 2224 (2018).

[Reply 3]

As pointed out by Reviewer#2, the large Seebeck effect originating from interfaces were reported in two dimensional electrons induced at oxide interfaces. In our experiments, however, we confirmed that the FeSe/SrTiO₃ interface does not have a large Seebeck coefficient. As shown in Figure 2, the large Seebeck coefficient was not observed without application of gate voltage ($V_G = 0$ V). If the FeSe/SrTiO₃ interface is the origin of the large Seebeck effect, the Seebeck coefficient should be large even at $V_G = 0$ V because the V_G does not reach the interface when the film is thicker than the screening length.

For further confirmation of the interface effect in ultrathin region, we have

performed additional experiments. We measured the Seebeck coefficient of the FeSe thin film on a KTaO_3 substrate, as discussed below (Reply 4). We found that the enhancement of the Seebeck effect occurred also on KTaO_3 substrate. This provides firm evidence that the enhanced Seebeck effect is not specific to SrTiO_3 substrate, suggesting that the FeSe/ SrTiO_3 interface is not a major origin.

Although the mechanism of the enhancement of the Seebeck effect in the gated FeSe thin film is different from that in the oxide heterointerfaces raised by Reviewer#2, those preceding studies are important for further understanding of the anomalous thermoelectric phenomena in two dimensional materials. In the revised manuscript, therefore, we have cited those papers as refs. 38-41 in the last paragraph in the Discussion session.

[Comment 4]

The best way to rule out the possibility of SrTiO_3 contribution, the authors can simply use MgO or LaAlO_3 single crystal as the substrate because MgO and LaAlO_3 are insulators do not contribute to the Seebeck coefficient. I am wondering why the authors did not use such insulating substrates though the authors used MgO substrate in the previous study [Shiogai, J. et al., Nature Phys. 12, 42 (2016)].

[Reply 4]

Thank you for your insightful comment. We agree that the Seebeck measurements in substrates other than SrTiO_3 is important. Although MgO and LaAlO_3 are raised as possible substrates by Reviewer#2, we have never tried to fabricate FeSe thin films on LaAlO_3 . It would take a long time to obtain the optimal conditions for the thin film fabrication by PLD. As for MgO, it is not appropriate for precise thermoelectric measurements because of the high thermal conductivity, which prohibits establishing temperature difference sufficient for reliable thermoelectric measurements, typically ~ 1 K or so.

Figure R3 Thermoelectric effect of FeSe thin film on KTaO₃ substrate.

a, Thermoelectric voltage ΔV under temperature difference ΔT . The measurements were performed at 200 K with $V_G = 4.3$ V. The ΔV dramatically increased with decreasing thickness d from ~ 14.5 nm. **b**, Thickness dependence of the Seebeck coefficient α . The α were estimated from the slope of the $\Delta V - \Delta T$ plot in **a** as $\alpha = -\Delta V / \Delta T$. FeSe/KTaO₃ exhibited enhancement of α similar to FeSe/SrTiO₃.

On this occasion to revise our manuscript, we have performed the Seebeck effect measurements on KTaO₃ substrate as an additional experiment since the growth condition of FeSe film was known in our previous study [J. Shiogai *et al.*, Phys. Rev. B **95**, 115101 (2017)]. As shown in Fig. R3, the FeSe-based electric double layer transistor fabricated on KTaO₃ shows the enhancement of the Seebeck effect with reducing the thickness. This is unambiguous evidence for the enhancement of the Seebeck effect in FeSe ultrathin films. Thus special contributions of SrTiO₃ substrate and the interface effect have been reasonably ruled out.

We have added the result of this additional experiments of FeSe/KTaO₃ in page 5 of Supplementary Information.

[Comment 5]

As the authors mentioned in the introduction part, low thermal conductivity is also required for the good thermoelectric materials. For the readers, it should be better if the authors comment on the thermal conductivity of FeSe.

[Reply 5]

The reason that we did not commented the thermal conductivity of FeSe is that the thermal conductivity measurements is impossible for our samples because the thickness of the film is too thin. However, as suggested by Reviewer#2, it is better to comment on the thermal conductivity to show a possible value of ZT .

In the revised manuscript, we added the description in page 9 that “Assuming the thermal conductivity for bulk Fe-based superconductors^{26,27}, $\kappa \sim 5 \text{ W m}^{-1} \text{ K}^{-1}$, the dimensionless figure of merit ZT of the FeSe ultrathin film reaches as large as ~ 1.5 at 280 K.”.

[Comment 6]

Some minor questions and comments

(Comment 6-1) In the abstract, the authors used S as the Seebeck coefficient whereas they used α in the main text.

(Comment 6-2) The size of the devices should be given in the Method section.

(Comment 6-3) The type of the thermocouple should be given in the Method section.

[Reply 6]

We appreciate these comments. We revised all points appropriately.

(Reply 6-1) In the revised manuscript, we use alfa α both in the abstract and main text.

(Reply 6-2) We have added the description on the size of the devices in the Methods section in the revised manuscript. The channel size is $1.2 \times 2 \text{ mm}^2$.

(Reply 6-3) We used type E Thermocouples. This information is also mentioned in the Methods section in the revised manuscript.

We sincerely hope that our replies have reasonably answered to all the comments.

Giant thermoelectric power factor in ultrathin FeSe superconductor

Sunao Shimizu¹, Junichi Shiogai², Nayuta Takemori¹, Shiro Sakai¹, Hiroaki Ikeda³, Ryotaro Arita¹, Tsutomu Nojima², Atsushi Tsukazaki², and Yoshihiro Iwasa^{*1,4}

¹RIKEN Center for Emergent Matter Science (CEMS), Wako, Saitama 351-0198, Japan

²Institute for Materials Research, Tohoku University, Sendai 980-8577, Japan

³Department of Physics, Ritsumeikan University, Kusatsu, Shiga 525-8577, Japan

⁴Quantum Phase Electronics Center (QPEC) and Department of Applied Physics,
University of Tokyo, Bunkyo, Tokyo 113-8656, Japan

The thermoelectric effect is attracting a renewed interest as a concept for energy harvesting technologies. Nanomaterials have been considered a key to realize efficient thermoelectric conversions owing to the low dimensional charge and phonon transports. In this regard, recently emerging two-dimensional (2D) materials could be promising candidates with novel thermoelectric functionalities. Here we report that FeSe ultrathin films, a high- T_c superconductor (T_c ; superconducting transition temperature), exhibit a superior thermoelectric responses. With decreasing thickness d , the electrical conductivity increases accompanying the emergence of high- T_c superconductivity; unexpectedly, the Seebeck coefficient α shows a concomitant increase as a result of the appearance of novel 2D natures. When d is reduced down to ~ 1 nm, the thermoelectric power factor at 50 K and room temperature reach unprecedented values as high as 13,000 and 260 $\mu\text{W cm}^{-1} \text{K}^{-2}$, respectively. The large thermoelectric effect in high T_c superconductors indicates the high potential of 2D layered materials towards multi-functionalization.

2D materials are expanding their arena in terms of richness in material type, properties, and functions, which range from electronic devices to catalysts and

medicines^{1,2}. Thermoelectric generation is one of the physical functions in which 2D materials are anticipated to be superior in comparison to their bulk counterparts. As shown in Fig. 1a, the density of states (DOS) in 2D semiconductors is considerably different from that of three-dimensional (3D) materials at the band edge singularity³. Since the Seebeck coefficient α is related to the profile of the DOS at the Fermi energy, 2D or low dimensional structures are considered to be advantageous for enhancing thermoelectric performance. Such a concept was proposed originally for semiconductor quantum wells and superlattices⁴; however, recently-emerging 2D layered materials provide naturally-formed atomic layers and their hetero-structures⁵, which are an ideal platform to elicit their intrinsic 2D nature. For characterization of thermoelectric properties of nanomaterials, on-chip device measurements have been often utilized⁶⁻⁸. Although the device configuration used for the measurements is not directly adapted to practical applications, it is highly powerful for realizing ideal conditions including the structures free from significant disorder and the tunable carrier density and thus for elucidating the intrinsic performance of materials. This method also fits the thermoelectric characterization of 2D materials in the present study.

The performance of thermoelectric semiconductors is measured by the figure of merit $ZT = \alpha^2 T / \rho \kappa$ (where ρ is the electrical resistivity, κ is the thermal conductivity,

and T is the absolute temperature). Therefore, materials with the large power factor α^2/ρ can be candidates for high ZT . In order to maximize α^2/ρ , we propose to extensively investigate recent 2D layered materials. In addition to the possible enhancement of the Seebeck effect in 2D DOS, an important characteristic of the recent 2D materials is their excellent crystallinity, which is preferable for keeping a large conductivity even in nano-thick monolayers.

For our purpose, 3d transition-metal-based compounds should be more favorable than 4d and 5d counterparts because the wave functions of 3d-based compounds are more localized, generally causing a larger effective mass m^* and thus the larger DOS (Figs. 1a and 1b). Among 3d based materials, we chose FeSe, firstly because a relatively large m^* ranging from 2 to 4 m_e has been reported in heavily electron-doped regions^{9,10}, where m_e is the free electron mass. The physical properties of ultrathin FeSe have attracted much attention because of the appearance of the unexpected high- T_c superconducting phase by reducing the film thickness down to a monolayer, the T_c of which reaches 65 K^{11,12} or 100 K¹³. Surprisingly, the high conductivity value survives even in monolayer FeSe^{11,14,15}; this is in stark contrast to conventional semiconductor thin films, where the resistance increases with reducing the thickness.

Here we report simultaneous measurements of α and ρ while controlling the thickness d of FeSe films on SrTiO₃ (001) substrates in an electric-double-layer transistor configuration, which is illustrated in Fig. 1c. In previous studies, we succeeded in optimization of α^2/ρ with controlling n through the gate bias V_G and applied this technique to various materials¹⁶ (see Methods). When V_G is applied at ~ 220 K, which is just above the glass transition temperature of the ionic liquid used in this study (see Methods), the cations or anions are self-aligned on the surface of FeSe (Fig. 1d); thus, charge carriers are electrostatically accumulated to form the electric double layer^{17,18}. On the other hand, when a certain level of V_G is applied at higher temperatures such as ~ 245 K or above, an electrochemical reaction takes place at the liquid-solid interface, and the topmost FeSe layer dissolves into the ionic liquid in a pseudo layer-by-layer manner¹⁵. Therefore, systematic investigation of the thermoelectric properties from bulk to ultrathin FeSe now becomes possible at a wide temperature range from 10 K to around room temperature. We found that the thermoelectric effect is dramatically enhanced with reducing d down to ~ 1 nm and thermoelectric power factor at 50 K and room temperature reach unprecedented values as high as 13,000 and 260 $\mu\text{W cm}^{-1} \text{K}^{-2}$, respectively. The coexistence of giant thermoelectric power factor and high- T_c superconductivity indicates the high potential of 2D layered materials towards

multi-functionalization. It should be noted that the parasitic conduction of the SrTiO₃ substrate is ruled out because the gate electric field on SrTiO₃ through the FeSe thin films is negligible in the present configuration due to the screening effect in the metallic conducting FeSe and also the formation of the Schottky barrier at the FeSe/SrTiO₃ interface¹⁹⁻²¹ (see Supplementary Figure 1 and Supplementary Note 1).

Results

Electrochemically enhanced Seebeck effect in FeSe thin film. Figure 2a shows the thermoelectric voltage ΔV as a function of the temperature difference ΔT between two thermocouples (see Fig. 1c) in Sample A at $V_G = 5$ V. The thermoelectric measurement was done at 200 K; possible conduction paths through the ionic liquid are completely eliminated. The device was cooled down to 200 K for the measurement at each thickness d after electrochemical etching at higher temperatures¹⁵. All the ΔV plots for different d 's linearly depended on ΔT , securing the accurate characterization of the Seebeck effect. At 200 K, ΔV was dramatically enhanced across the sign change with decreasing d from ~ 18 nm to ~ 1 nm. Figure 2b plots the d dependence of $\alpha (= -\Delta V/\Delta T)$ at 200 K for $V_G = 5$ V. A very small α of $+3.8 \mu\text{V K}^{-1}$ at $d \sim 18$ nm is consistent with the reported values on bulk FeSe^{22,23}, where such a small α value reflects the semimetallic electronic structure²⁴. With

the thickness reduction, on the other hand, the absolute value of α was surprisingly enhanced by two orders of magnitude up to $|-454 \mu\text{V K}^{-1}|$. Importantly, this giant Seebeck effect is observed only under gate bias; as seen in the main panel of Fig. 2b (gray circles), the large α obtained at $V_G = 5 \text{ V}$ was suppressed to bulk-like small values by switching off V_G to 0 V . This definitely rules out the existence of an oxygen deficient layer at the surface of SrTiO_3 substrate as a source of the large Seebeck response. Also, it is emphasized that, the high- T_c superconductivity appears by applying $V_G = 5 \text{ V}$ and disappears by removing V_G ^{15,17,18}, as shown in the inset of Fig. 2b. The simultaneous emergence of the giant thermoelectric response and the high- T_c superconductivity proves that these two novel transport properties arise from the same electronic state of FeSe thin films.

Another noticeable feature of FeSe thin films is the low electrical resistance realized even in ultrathin regions. Figure 2c shows the two dimensional sheet resistance ρ_{2D} of Sample A for $V_G = 0 \text{ V}$ (gray circles) and 5 V (blue circles) as a function of d . When starting from the initial state with $d \sim 18 \text{ nm}$, the two dimensional resistivity at 200 K , $\rho_{2D}^{200 \text{ K}}$, first increased with decreasing d for both $V_G = 0 \text{ V}$ and 5 V , as expected^{15,25}. With further decreasing d , $\rho_{2D}^{200 \text{ K}}$ at $V_G = 5 \text{ V}$ showed a small peak at around $d \sim 11 \text{ nm}$ and kept small values down to $d \sim 1 \text{ nm}$ because the gated topmost layer of FeSe and the

charge transfer layer at the FeSe/SrTiO₃ interface dominate the electrical transport of the thin film (see Supplementary Figure 2 and Supplementary Note 2 for the details of the d dependence of $\rho_{2D}^{200\text{ K}}$). Such a low electrical resistance irrespective of the film thickness is consistent with the previous studies; for example, the resistivity of monolayer or few layer MBE-grown FeSe^{11,14} is comparable to that of 10 nm thick (~15 layers) FeSe^{17,18} owing to the interface or surface electron doping. Actually, $\rho_{2D}^{200\text{ K}} \sim 1\text{ k}\Omega$ at $V_G = 5\text{ V}$ in the thin limit (Fig. 2c) is close to that in doped FeSe monolayers^{11,14,15}. On the other hand, the small α and high $\rho_{2D}^{200\text{ K}}$ at $V_G = 0\text{ V}$ indicate that the charge transfer layer does not produce the enhanced values of α . Consequently, the thermoelectric power factor α^2/ρ at 200 K achieved a dramatic development in Fig. 2d owing to the concomitant enhancement of α and the electrical resistivity $\rho = \rho_{2D} \times d$, which rarely occurs in the framework of conventional material design and fabrication. Along with the reduction of d from 18 nm to 1 nm, α^2/ρ kept increasing and finally reached $\sim 1,500\ \mu\text{W cm}^{-1}\text{ K}^{-2}$.

Temperature-thickness mapping of thermoelectric response in FeSe. Figures 3a and 3b display the temperature T – thickness d mappings of the absolute value of α (i.e., $|\alpha|$) and α^2/ρ , respectively, for another FeSe thin film, Sample B. The values of $|\alpha|$ and α^2/ρ showed dramatic developments in the nanometer-thick region, which agrees well

with the results for Sample A (see Figs. 2b and 2c). Moreover, the enhancement for both $|\alpha|$ and α^2/ρ covers a wide temperature range from 50 K (just above T_c) to 280 K. Figure 3c summarizes α^2/ρ for representative thermoelectric materials that possess high α^2/ρ values (see Supplementary Table 1). The values of α^2/ρ for the FeSe ultrathin film increased from $\sim 260 \mu\text{W cm}^{-1} \text{K}^{-2}$ at 280 K up to $\sim 13,000 \mu\text{W cm}^{-1} \text{K}^{-2}$ at 50 K, being the largest among existing bulk materials reported so far. Assuming the thermal conductivity for bulk Fe-based superconductors^{26,27}, $\kappa \sim 5 \text{ W m}^{-1} \text{K}^{-1}$, the dimensionless figure of merit ZT of the FeSe ultrathin film reaches as large as ~ 1.5 at 280 K.

Common trend of Seebeck effect in Fe-based high- T_c superconductors. The detailed temperature dependence of α for different d 's is presented in Fig. 4a to show the unusual thermoelectric response in FeSe. Except for the initial thickness (19.1 nm) with moderate temperature dependence, α for Sample B showed a peak at around ~ 200 K, which follows neither the T -linear behavior expected in conventional metals nor the phonon drag thermopower (see Supplementary Figure 1 and Supplementary Note 1). Actually, the temperature dependence of α in the FeSe thin film is qualitatively similar

to that in bulk Fe-based high- T_c superconductors such as $\text{Ba}(\text{Fe}_{0.9}\text{Co}_{0.1})_2\text{As}_2$ ²⁸, $\text{LaFeAs}(\text{O}_{0.9}\text{F}_{0.1})$ ²⁹, and $\text{La}(\text{Fe}_{0.9}\text{Co}_{0.1})\text{AsO}$ ³⁰, as shown in Fig. 4b. This trend can be seen even more clearly in Fig. 4c, where the data in Figs. 4a and 4b are normalized by the peak value α^{peak} of each sample. These similarities further prove that α observed in Fig. 4a is attributed to FeSe itself rather than other artifacts such as substrates and ionic liquids. The characteristic temperature dependence of $\alpha/\alpha^{\text{peak}}$ in Fig. 4c is considered unique to Fe-based superconductors and has been discussed in the context of quantum criticality^{31–33} or the two carrier model^{28,34}. For example, it was reported that $|\alpha|/T$ in $\text{Ba}(\text{Fe}_{0.9}\text{Co}_{0.1})_2\text{As}_2$ ²⁸ shows a divergence above T_c and a strong enhancement when in proximity to the quantum critical point.

Discussion

The transfer characteristics (the V_G dependence of resistance) of a FeSe thin film (see Supplementary Figure 3 and Supplementary Note 3) indicates that the dominant carriers change from holes to electrons with reducing the thickness. This behavior is consistent with the band structure evolution derived from the angle-resolved photoemission spectroscopy (ARPES)^{12,35–37}, which clarified that the hole pocket at the Γ -point

disappears, and a gap of ~ 60 meV is opened at the M-point³⁵ in the electron-doped monolayer FeSe. This band structure of FeSe monolayer should be beneficial for the enhancement of $|\alpha|$. We calculated the Seebeck coefficient for the undoped bulk FeSe and electron-doped monolayer FeSe at $T = 280$ K (see Supplementary Figure 4 and Supplementary Note 4), and obtained α values as $+5 \mu\text{V K}^{-1}$ and $-200 \mu\text{V K}^{-1}$ for the bulk and the monolayer FeSe, respectively. These estimations reasonably explain the experimental values of α at 280 K: $+17 \mu\text{V K}^{-1}$ and $-245 \mu\text{V K}^{-1}$ (Fig. 3A) for the initial ($d \sim 19.1$ nm) and final ($d \sim 1$ nm) thicknesses, respectively. On the other hand, our calculation based on the Fermi liquid picture predicts T -linear behavior and does not explain the non-monotonous temperature dependence of α in ultrathin FeSe. The experimentally observed broad peak in α located at around 200 K (Fig. 4a) is suggestive of a crucial role of electronic correlations in the Seebeck response of ultrathin FeSe; in fact, the recent ARPES studies pointed out a strong electronic correlation⁹ in the high- T_c phase of FeSe. A quantitative theoretical analysis of this effect remains to be performed.

Nanostructures or low-dimensional structures have been a powerful guideline for the exploration of high performance thermoelectric materials^{8,38-41}. The present results show that further enhancement of thermoelectric properties should be possible, if peculiar band structures of nano-structured systems including 2D layered materials are

combined with additional ingredients such as strong electronic correlations. The unprecedented coexistence of giant thermoelectric power factor and high- T_c superconductivity in ultrathin FeSe exemplifies that there may exist unknown multifunctional materials waiting to be disclosed in extreme conditions, illuminating a new research direction of functional thermoelectric materials.

Methods

Device fabrication. We fabricated ion-gated devices based on FeSe thin films on SrTiO₃ substrates¹⁵ with channel size of $1.2 \times 2 \text{ mm}^2$. The details of the thin film preparation were reported in our previous study¹⁵. The device structure used in this study is schematically shown in Fig. 1c. The FeSe thin films were patterned by using a laser cutter to perform four terminal resistance measurements. The gold wires were attached at both edges of the patterned film, working as a drain terminal D and a source terminal S . An ionic liquid, which worked as a gate dielectric, was placed on the FeSe surface. We used N,N -diethyl- N -(2-methoxyethyl)- N -methylammonium bis-(trifluoromethylsulfonyl)-imide (DEME-TFSI) as the ionic liquid. A Pt plate was placed on top of them, working as a gate electrode.

Thermoelectric measurements under gate biases. As shown in Fig. 1c, a heater and

a heat sink were attached to either side of the ion-gated device to produce a thermal gradient. The type E thermocouples were attached to monitor the temperature difference ΔT and the thermoelectric voltage ΔV . The thermocouples were also used for the four-terminal resistance measurements. The temperature difference ΔT (0 to 1 K) and the voltage ΔV between the thermocouples were measured, and the values of α were evaluated from the slope of the ΔV - ΔT plots (See Fig. 2a). This device configuration allows us to measure α and σ simultaneously. The thermoelectric measurements with solid⁴²⁻⁴⁸ and ionic gate dielectrics⁴⁹⁻⁵⁸ is widely accepted as a method to evaluate the thermoelectric properties of semiconductors with changing the carrier densities.

Calculations. We performed first-principles band structure calculations using the Perdew-Burke-Ernzerhof parameterization of the generalised gradient approximation⁵⁹ and the full-potential (linearized) augmented plane-wave method, with the inclusion of spin-orbit coupling as implemented in the wien2k code⁶⁰. Muffin-tin radii (R_{MT}) of 2.38 and 2.11 Bohr were used for Fe and Se, respectively. The maximum modulus for the reciprocal vectors K_{max} was chosen such that $R_{\text{MT}}K_{\text{max}} = 7.0$ and a $10 \times 10 \times 10$ k-mesh in the first Brillouin zone was used. The tight-binding Hamiltonian for the $3d$ orbitals of the Fe atom was constructed with Wannier90⁶¹ and wien2wannier⁶².

Data availability. The authors declare that all data supporting the findings of this

study are available within the paper and its Supplementary Information.

References

1. Fiori, G. *et al.* Electronics based on two-dimensional materials. *Nature Nanotechnol.* **9**, 768–779 (2014).
2. Chhowalla, M. *et al.* The chemistry of two-dimensional layered transition metal dichalcogenide nanosheets. *Nature Chem.* **5**, 263–275 (2013).
3. Dresselhaus, M. S. *et al.* New Directions for Low-Dimensional Thermoelectric Materials. *Adv. Mater.* **19**, 1043–1053 (2007).
4. Hicks, L. D. & Dresselhaus, M. S. Effect of quantum-well structures on the thermoelectric figure of merit. *Phys. Rev. B* **47**, 12727–12731 (1993).
5. Geim, A. K. & Grigorieva, I. V. Van der Waals heterostructures. *Nature* **499**, 419–425 (2013).
6. Venkatasubramanian, R., Siivola, E., Colpitts, T. & O’Quinn, B. Thin-film thermoelectric devices with high room-temperature figures of merit. *Nature* **413**,

- 597–602 (2001).
7. Harman, T. C., Taylor, P. J., Walsh, M. P. & La Forge, B. E. Quantum dot superlattice thermoelectric materials and devices. *Science* **297**, 2229–2232 (2002).
 8. Ohta, H. *et al.* Giant thermoelectric Seebeck coefficient of a two-dimensional electron gas in SrTiO₃. *Nature Mater.* **6**, 129–134 (2007).
 9. Wen, C. H. P. *et al.* Anomalous correlation effects and unique phase diagram of electron-doped FeSe revealed by photoemission spectroscopy. *Nature Commun.* **7**, 10840 (2016).
 10. Seo, J. J. *et al.* Superconductivity below 20 K in heavily electron-doped surface layer of FeSe bulk crystal. *Nature Commun.* **7**, 11116 (2016).
 11. Wang, Q.-Y. *et al.* Interface-induced high-temperature superconductivity in single unit-cell FeSe films on SrTiO₃. *Chinese Phys. Lett.* **29**, 037402 (2012).
 12. Liu, D. *et al.* Electronic origin of high-temperature superconductivity in single-layer FeSe superconductor. *Nature Commun.* **3**, 931 (2012).
 13. Ge, J.-F. *et al.* Superconductivity above 100 K in single-layer FeSe films on doped SrTiO₃. *Nature Mater.* **14**, 285–289 (2014).

14. Sun, Y. *et al.* High temperature superconducting FeSe films on SrTiO₃ substrates. *Sci. Rep.* **4**, 6040 (2014).
15. Shiogai, J., Ito, Y., Mitsuhashi, T., Nojima, T. & Tsukazaki, A.
Electric-field-induced superconductivity in electrochemically etched ultrathin FeSe films on SrTiO₃ and MgO. *Nature Phys.* **12**, 42–46 (2016).
16. Bisri, S. Z., Shimizu, S., Nakano, M. & Iwasa, Y. Endeavor of Iontronics: From Fundamentals to Applications of Ion-Controlled Electronics. *Adv. Mater.* **29**, 1607054 (2017).
17. Lei, B. *et al.* Evolution of High-Temperature Superconductivity from a Low T_c Phase Tuned by Carrier Concentration in FeSe Thin Flakes. *Phys. Rev. Lett.* **116**, 077002 (2016).
18. Hanzawa, K., Sato, H., Hiramatsu, H., Kamiya, T. & Hosono, H. Electric field-induced superconducting transition of insulating FeSe thin film at 35 K. *Proc. Natl. Acad. Sci. U. S. A.* **113**, 3986–3990 (2016).
19. Wu, C. T. *et al.* Heterojunction of Fe(Se_{1-x}Te_x) superconductor on Nb-doped SrTiO₃. *Appl. Phys. Lett.* **96**, 122506 (2010).
20. Zhang, W. *et al.* Interface charge doping effects on superconductivity of

- single-unit-cell FeSe films on SrTiO₃ substrates. *Phys. Rev. B* **89**, 060506(R) (2014).
21. Zhang, H. *et al.* Origin of charge transfer and enhanced electron-phonon coupling in single unit-cell FeSe films on SrTiO₃. *Nature Commun.* **8**, 214 (2017).
 22. McQueen, T. M. *et al.* Extreme sensitivity of superconductivity to stoichiometry in Fe_{1+δ}Se. *Phys. Rev. B* **79**, 014522 (2009).
 23. Song, Y. J. *et al.* Superconducting Properties of a Stoichiometric FeSe Compound and Two Anomalous Features in the Normal State. *J. Korean Phys. Soc.* **59**, 312–316 (2011).
 24. Nakayama, K. *et al.* Reconstruction of band structure induced by electronic nematicity in an FeSe superconductor. *Phys. Rev. Lett.* **113**, 237001 (2014).
 25. Schneider, R., Zaitsev, A. G., Fuchs, D. & Löhneysen, H. V. Superconductor-insulator quantum phase transition in disordered FeSe thin films. *Phys. Rev. Lett.* **108**, 257003 (2012).
 26. Machida, Y. *et al.* Possible sign-reversing s-wave superconductivity in co-doped BaFe₂As₂ proved by thermal transport measurements. *J. Phys. Soc. Japan* **78**, 073705 (2009).

27. Checkelsky, J. G. *et al.* Thermal hall conductivity as a probe of gap structure in multiband superconductors: The case of $\text{Ba}_{1-x}\text{KxFe}_2\text{As}_2$. *Phys. Rev. B* **86**, 180502(R) (2012).
28. Arsenijević, S. *et al.* Pressure effects on the transport coefficients of $\text{Ba}(\text{Fe}_{1-x}\text{Co}_x)_2\text{As}_2$. *Phys. Rev. B* **84**, 075148 (2011).
29. Zhu, Z. W. *et al.* Nernst effect of a new iron-based superconductor $\text{LaO}_{1-x}\text{F}_x\text{FeAs}$. *New J. Phys.* **10**, 063021 (2008).
30. Kondrat, A., Behr, G., Büchner, B. & Hess, C. Unusual Nernst effect and spin density wave precursors in superconducting $\text{LaFeAsO}_{1-x}\text{F}_x$. *Phys. Rev. B* **83**, 092507 (2011).
31. Gooch, M., Lv, B., Lorenz, B., Guloy, A. M. & Chu, C. W. Critical scaling of transport properties in the phase diagram of iron pnictide superconductors $\text{K}_x\text{Sr}_{1-x}\text{Fe}_2\text{As}_2$ and $\text{K}_x\text{Ba}_{1-x}\text{Fe}_2\text{As}_2$. *J. Appl. Phys.* **107**, 09E145 (2010).
32. Maiwald, J., Jeevan, H. S. & Gegenwart, P. Signatures of quantum criticality in hole-doped and chemically pressurized EuFe_2As_2 single crystals. *Phys. Rev. B* **85**, 024511 (2012).
33. Arsenijević, S. *et al.* Signatures of quantum criticality in the thermopower of

- Ba(Fe_{1-x}Co_x)₂As₂. *Phys. Rev. B* **87**, 224508 (2013).
34. Sales, B. C., McGuire, M. A., Sefat, A. S. & Mandrus, D. A semimetal model of the normal state magnetic susceptibility and transport properties of Ba(Fe_{1-x}Co_x)₂As₂. *Phys. C* **470**, 304–308 (2010).
35. He, S. *et al.* Phase diagram and electronic indication of high-temperature superconductivity at 65 K in single-layer FeSe films. *Nature Mater.* **12**, 605–610 (2013).
36. He, J. *et al.* Electronic Evidence of an Insulator-Superconductor Transition in Single-Layer FeSe/SrTiO₃ Films. *Proc. Natl. Acad. Sci. U. S. A.* **111**, 18501–18506 (2014).
37. Miyata, Y., Nakayama, K., Sugawara, K., Sato, T. & Takahashi, T. High-temperature superconductivity in potassium-coated multilayer FeSe thin films. *Nature Mater.* **14**, 775–779 (2015).
38. Choi, W. S., Ohta, H., Moon, S. J., Lee, Y. S. & Noh, T. W. Dimensional crossover of polaron dynamics in Nb:SrTiO₃/SrTiO₃ superlattices: Possible mechanism of thermopower enhancement. *Phys. Rev. B* **82**, 024301 (2010).
39. Ohta, H. *et al.* Field-induced water electrolysis switches an oxide semiconductor

- from an insulator to a metal. *Nature Commun.* **1**, 118 (2010).
40. Ohta, H. *et al.* Unusually large enhancement of thermopower in an electric field induced two-dimensional electron gas. *Adv. Mater.* **24**, 740–744 (2012).
 41. Zhang, Y. *et al.* Double thermoelectric power factor of a 2D electron system. *Nature Commun.* **9**, 2224 (2018).
 42. Zavaritsky, N. V. Phonon drag in two-dimensional electron systems. *Phys. B+C* **126**, 369–376 (1984).
 43. Gallagher, B. L., Gibbins, C. J., Pepper, M. & Cantrell, D. G. The thermopower of Si inversion layers. *Semicond. Sci. Technol.* 456–459 (1987).
 44. Small, J., Perez, K. & Kim, P. Modulation of Thermoelectric Power of Individual Carbon Nanotubes. *Phys. Rev. Lett.* **91**, 256801 (2003).
 45. Yoshikawa, A. *et al.* Electric-Field Modulation of Thermopower for the KTaO_3 Field-Effect Transistors. *Appl. Phys. Express* **2**, 121103 (2009).
 46. Checkelsky, J. & Ong, N. Thermopower and Nernst effect in graphene in a magnetic field. *Phys. Rev. B* **80**, 081413 (2009).
 47. Zuev, Y., Chang, W. & Kim, P. Thermoelectric and Magnetothermoelectric Transport Measurements of Graphene. *Phys. Rev. Lett.* **102**, 096807 (2009).

48. Ohta, H. *et al.* Field-modulated thermopower in SrTiO₃-based field-effect transistors with amorphous 12CaO · 7Al₂O₃ glass gate insulator. *Appl. Phys. Lett.* **95**, 113505 (2009).
49. Shimizu, S., Ono, S., Hatano, T., Iwasa, Y. & Tokura, Y. Enhanced cryogenic thermopower in SrTiO₃ by ionic gating. *Phys. Rev. B* **92**, 165304 (2015).
50. Yoshida, M. *et al.* Gate-Optimized Thermoelectric Power Factor in Ultrathin WSe₂ Single Crystals. *Nano Lett.* **16**, 2061–2065 (2016).
51. Shimizu, S. *et al.* Enhanced thermopower in ZnO two-dimensional electron gas. *Proc. Natl. Acad. Sci. U. S. A.* **113**, 6438–6443 (2016).
52. Shimizu, S. *et al.* Thermoelectric Detection of Multi-Subband Density of States in Semiconducting and Metallic Single-Walled Carbon Nanotubes. *Small* **12**, 3388–3392 (2016).
53. Yanagi, K. *et al.* Tuning of the Thermoelectric Properties of One-Dimensional Materials Networks by Electric Double Layer Techniques Using Ionic Liquids. *Nano Lett.* **14**, 6437 (2014).
54. Takayanagi, R., Fujii, T. & Asamitsu, A. Simultaneous control of thermoelectric properties in p- and n-type materials by electric double-layer gating: New design

- for thermoelectric device. *Appl. Phys. Express* **8**, 051101 (2015).
55. Chien, Y.-Y., Yuan, H. T., Wang, C.-R. & Lee, W.-L. Thermoelectric Power in Bilayer Graphene Device with Ionic Liquid Gating. *Sci. Rep.* **6**, 20402 (2016).
56. Kawasugi, Y. *et al.* Simultaneous enhancement of conductivity and Seebeck coefficient in an organic Mott transistor. *Appl. Phys. Lett.* **109**, 233301 (2016).
57. Pu, J. *et al.* Enhanced thermoelectric power in two-dimensional transition metal dichalcogenide monolayers. *Phys. Rev. B* **94**, 014312 (2016).
58. Kawai, H. *et al.* Thermoelectric properties of WS₂ nanotube networks. *Appl. Phys. Express* **10**, 015001 (2017).
59. Perdew, J. P., Bruke, K. & Ernzerhof, M. Generalized Gradient Approximation Made Simple. *Phys. Rev. Lett.* **77**, 3865–3868 (1996).
60. Blaha, P., Schwarz, K., Madsen, G., Kvasnicka, D. & Luitz, J. *WIEN2k, An Augmented Plane Wave + Local Orbitals Program for Calculating Crystal Properties.* (Tech. Univ. Wien, 2001).
61. Mostofi, A. A. *et al.* wannier90: A tool for obtaining maximally-localised Wannier functions. *Comput. Phys. Commun.* **178**, 685–699 (2008).
62. Kuneš, J. *et al.* Wien2wannier: From linearized augmented plane waves to

maximally localized Wannier functions. *Comput. Phys. Commun.* **181**, 1888–1895 (2010).

Acknowledgments

This work was supported by JSPS KAKENHI Grant Numbers JP25000003, JP16H00923 (SATL), JP16H06345, JP17H02928, JP17K19060.

Author Contributions

S.Sh. and J.S. equally contributed to this work. S.Sh., J.S., T.N., A.T., and Y.I. conceived and designed this work. S.Sh., N.T., S.Sa., A.T., and Y.I. wrote the paper. S.Sh. and J.S. performed all the measurements, and N.T., S.Sa., H.I., and R.A. conducted the calculations. All authors contributed to discussions.

Additional information

Supplementary Information is available in the online version of the paper. Reprints and permissions information is available at www.nature.com/reprints. The authors declare no competing financial interests. Correspondence and requests for materials should be addressed to Y.I. (iwasa@ap.t.u-tokyo.ac.jp).

Figures and Figure legends

Figure 1 Schematic structure of FeSe thin film device for thermoelectric measurement. **a**, Schematic illustration of the electronic density of states (DOS) for three-dimensional (3D) and two-dimensional (2D) electrons. **b**, The large effective mass m^* enhances the DOS, which is favorable to the enhancement of the Seebeck effect. **c**, Device structure for thermoelectric measurement. V_{SD} and V_G stand for the source (S) – drain (D) voltage and the gate bias voltage, respectively. When V_G is applied to the Pt plate, ions in the ionic liquid are redistributed, forming an electric double layer on the surface of the FeSe film. **d**, Enlarged illustration of the ionic liquid/FeSe interface. Under the positive gate

bias, *N,N*-diethyl-*N*-(2-methoxyethyl)-*N*-methylammonium cations, DEME+, align on the surface of FeSe. The thickness d of the FeSe thin film was tuned by electrochemical etching¹⁵. See Methods for details of the device structure and fabrication.

Figure 2 Thinning-induced enhancement of thermoelectric effect in FeSe

thin film. a, Thermoelectric voltage ΔV under temperature difference ΔT in an

FeSe thin film, Sample A. The thermoelectric measurements were performed at

200 K with $V_G = 5$ V. The value of ΔV changed its sign and dramatically increased with decreasing thickness d from ~ 18 nm to ~ 1 nm. **b**, Thickness d dependence of the Seebeck coefficient α . The values of α were estimated from the slope of the $\Delta V - \Delta T$ plot in **a** as $\alpha = -\Delta V / \Delta T$. The inset shows the temperature T dependence of the normalized sheet resistance R_s with respect to 200 K. Superconductivity appeared when $V_G = 5$ V was applied. **c**, Variation of sheet resistance $R_s^{200\text{ K}}$ at 200 K as a function of d . The value of $R_s^{200\text{ K}}$ for $V_G = 5$ V showed a weak d dependence (blue circles), whereas that for $V_G = 0$ V increased with decreasing d (gray circles). **d**, Thickness d dependence of thermoelectric power factor $\alpha^2 \sigma$ exhibiting anomalous enhancement in the ultrathin limit. The value of $\alpha^2 \sigma$ at 200 K in the thick region is comparable to that in bulk FeSe²², while $\alpha^2 \sigma$ increased with decreasing d because of the double-digit increase of α in ultrathin regions in **b**. The error bar of d corresponds to $\sim \pm 0.5$ nm, which was estimated from the surface roughness of the initial thin film¹⁵.

Figure 3 Giant thermoelectric response in ultrathin FeSe. **a**, Mapping of Seebeck coefficient α of FeSe (Sample B) against temperature T and thickness d . Here, the absolute value of α (i.g., $|\alpha|$) for $V_G = 3.95$ V was plotted. The white circles correspond to the onset temperature T_c of the superconducting transition. **b**, Evolution of thermoelectric power factor $\alpha^2 \sigma$ of FeSe (Sample B) above T_c .

The value of $\alpha^2\sigma$ increased with decreasing d , mainly due to the large enhancement of α shown in **a. c**, Comparison of temperature dependence of $\alpha^2\sigma$ among representative thermoelectric materials. The values of $\alpha^2\sigma$ in the FeSe ultrathin film were larger than any existing bulk materials reported so far in a wide temperature range (see Supplementary Table 1 for the reference of the experimental values). The data point for Sample A at 200 K shows fair reproducibility.

Figure 4 Comparison of Seebeck effect between FeSe thin film and bulk Fe-based superconductors. **a**, Seebeck coefficient α vs. temperature T in FeSe thin film (Sample B). The values of α were enhanced by the thickness reduction in all temperature regions. **b**, α - T curves in representative Fe-based high- T_c superconductors. $\text{Ba}(\text{Fe}_{0.9}\text{Co}_{0.1})_2\text{As}_2$ ²⁸, $\text{LaFeAs}(\text{O}_{0.9}\text{F}_{0.1})$ ²⁹, and $\text{La}(\text{Fe}_{0.9}\text{Co}_{0.1})\text{AsO}$ ³⁰ show the peak behavior at around 150 K, in a similar manner to the FeSe thin film in **a**. **c**, Temperature dependence of α normalized by the peak value α^{peak} for FeSe thin films and bulk materials in **b**. Overall temperature variation of α is seemingly common for Fe-based superconductors.

Supplementary Information for “Giant thermoelectric power factor in ultrathin FeSe superconductor”

Supplementary Note 1: Evaluation of parasitic conduction on SrTiO₃ surface

Here we provide five pieces of evidence that there is no contribution from the SrTiO₃ substrates to the Seebeck effect and the electrical conductivity in FeSe ultrathin films.

S1-1) Negligible conductance of SrTiO₃ substrate under gating

It has been reported by many groups that an oxide insulator, SrTiO₃, shows metallic conduction with ionic liquid gating¹⁻⁷. In our present study, the FeSe thin films were fabricated on SrTiO₃ substrates. First of all, therefore, we would like to rule out the possibility that the charge carriers are unintentionally induced at the surface of the SrTiO₃ substrates under the application of the gate bias voltage V_G .

We prepared two configurations for the test of the ionic liquid gating on SrTiO₃, as shown in Supplementary Fig. 1. Supplementary Fig. 1a is a conventional electric-double-layer transistor configuration (Configuration A) with an insulating SrTiO₃ channel and Ti/Au Ohmic contact electrodes. The surface of SrTiO₃ just beneath the Ti/Au contact electrodes was exposed to the Ar-ion beam in order to make oxygen deficient SrTiO_{3- δ} layer⁸. This process forms the Ohmic contact between the Ti/Au

contact electrodes and the SrTiO₃ channel^{1-7,9}. Supplementary Fig. 1b is also an electric-double-layer transistor configuration (Configuration B), where FeSe thin films are used as contact electrodes. This configuration mimics a possible hole opening of FeSe films due to the non-uniform electrochemical etching. We measured the transfer characteristics (the drain-source current I_D versus V_G) for the two configurations at 220 K and found strikingly contrasting results. As seen in Supplementary Fig. 1c, I_D in Configuration A increased with increasing V_G , exhibiting a typical n -type field effect transistor operation in SrTiO₃^{1-4,6,7}. On the other hand, I_D in Configuration B did not show any enhancement under the application of V_G . This is due to the formation of the Schottky barrier between FeSe and insulating SrTiO₃, which is consistent with the charge transfer picture¹⁰⁻¹². Thus, when FeSe thin film is gated with ionic liquid, the charge carriers are not induced at the surface of SrTiO₃ substrates.

S1-2) Estimation of Seebeck coefficient assuming conductive SrTiO₃ surface

There still remains a possibility that Configuration B in Supplementary Fig. 1b is not an exact copy of the real experiments. Here, we assume that the gate electric field accumulates some amount of charge carriers in the SrTiO₃ substrate and that both FeSe and the SrTiO₃ substrate contribute to the observed Seebeck coefficient in Fig. 2b in the main text. In this case, they form a parallel circuit and the total Seebeck coefficient α_{total} is written as

$$\alpha_{\text{total}} = \frac{\frac{1}{\rho_{2\text{D,FeSe}}} \times \alpha_{\text{FeSe}} + \frac{1}{\rho_{2\text{D,STO}}} \times \alpha_{\text{STO}}}{\frac{1}{\rho_{2\text{D,FeSe}}} + \frac{1}{\rho_{2\text{D,STO}}}}. \quad (\text{S1})$$

Here, the Seebeck coefficient and the two dimensional electrical resistivity for FeSe are α_{FeSe} and $\rho_{2\text{D,FeSe}}$, and those for the gated SrTiO₃ substrate are α_{STO} and $\rho_{2\text{D,STO}}$. We tentatively assume that the Seebeck effect of FeSe is constant at $\alpha_{\text{FeSe}} = +3.8 \mu\text{V/K}$ at 200 K (see Figs. 2a and 2b in the main text) against d . According to the literatures on FeSe thin flims^{13–16}, the two dimensional resistivity of the monolayer or few-layer FeSe on SrTiO₃ substrates, $\rho_{2\text{D,FeSe}}$, is $\sim 1 \text{ k}\Omega$. Importantly, the Seebeck effect of SrTiO₃ single crystals gated with ionic liquid was already reported, the values of $\rho_{2\text{D,STO}}$ and α_{STO} at 200 K are known, as summarized in Supplementary Fig. 1d. Substituting the values of $\rho_{2\text{D,FeSe}}$, α_{FeSe} , $\rho_{2\text{D,STO}}$, and α_{STO} into Eq. S1, α_{total} is estimated as listed in Supplementary Fig. 1d. The estimated values of α_{total} were $-50 \mu\text{V/K}$ at most even if the SrTiO₃ substrate acquired a highly metallic conduction. These considerations clearly exclude the possible parallel conduction of SrTiO₃ substrate and FeSe.

S1-3) Temperature dependence of Seebeck coefficient in doped SrTiO₃

Here we assume that, as extreme case, the observed Seebeck coefficient is all attributed to the SrTiO₃ substrate. In this case, however, the temperature dependence of the observed Seebeck effect in FeSe on SrTiO₃ (Sample B in the main text) is totally different from that in conducting SrTiO₃, as shown in Supplementary Fig. 1e. The values of α in FeSe thin films has a peak at around 150 to 200 K, which is typical behavior in Fe-based high- T_c superconductors^{17–19}, as shown in Fig. 4c; on the other

hand, α in conducting SrTiO₃ keeps increasing^{20,21} up to 1100 K (the data above 600 K is not shown). These results clearly demonstrate that the enhanced Seebeck effect in Fig. 2b stems from the high- T_c phase of FeSe.

S1-4) Phonon drag effect in thin films on SrTiO₃ substrates

The above three arguments rule out the possible electrical conduction of the SrTiO₃ substrate and its contribution to the enhanced Seebeck coefficient observed in the FeSe thin film. In this sub-section we show that the observed large thermopower is also irrelevant to the phonon-drag effect that is induced by the phonons in SrTiO₃ substrate.

When the thickness of thin films is several nanometers, the interaction between charge carriers in thin films and phonons in insulating substrates could induce the phonon drag effect at low temperatures where the thermal conductivity κ of the substrates is maximum²². As for SrTiO₃ single crystals, κ shows a sharp peak at around 20 K²³. However, α of FeSe thin films was zero at ~ 20 K, as shown in Supplementary Fig. 1e, because FeSe thin films become superconducting below ~ 40 K. In addition, the intensity of the phonon drag effect is inversely proportional to the fourth power of temperature, so that it should decay rapidly with increasing temperature and be pronounced only at around the peak temperature of κ of the substrates²². Thus, the effect of phonon drag triggered by SrTiO₃ substrate is unambiguously excluded.

S1-5) Seebeck effect of FeSe/KTaO₃

In addition to above arguments, a direct way to examine the contribution of SrTiO₃ substrates for the enhanced Seebeck effect is to measure the Seebeck effect of FeSe thin films fabricated on other substrates. Here we apply the ionic gating on FeSe/KTaO₃ to evaluate the thickness dependence of the Seebeck effect at 200 K.

Supplementary Figure 2a shows the thermoelectric voltage ΔV as a function of the temperature difference ΔT between two thermocouples (see Fig. 1c) on a FeSe thin film on a KTaO₃ substrate (Sample C) at $V_G = 4.3$ V. The thermoelectric measurement was done at 200 K to compare with the data for FeSe/SrTiO₃ in Figs. 2a and 2b. All the ΔV plots for different d 's linearly depended on ΔT , securing the accurate characterization of the Seebeck effect. The values of ΔV were dramatically enhanced with decreasing d , showing that the Seebeck effect of FeSe thin films is actually enhanced by reducing d even on KTaO₃ substrates.

Supplementary Figure 2b plots the d dependence of $\alpha (= -\Delta V / \Delta T)$ for Sample C at 200 K (red circles). With reducing the thickness from 14.5 nm, the absolute value of α was dramatically enhanced. The data for Sample A of FeSe/SrTiO₃, which is identical with that in Fig. 2b, are also plotted in Supplementary Fig. 2b. The values of α for both KTaO₃ and SrTiO₃ substrates show similar thickness dependence, suggesting that the enhanced Seebeck effect of FeSe does not depend on substrate materials.

Supplementary Note 2: Evolution of Seebeck effect with decreasing film thickness of FeSe

As reported by ARPES studies^{24–27} and discussed in the later section (Supplementary Note 3), the electric field carrier doping performed in this study should cause a modulation of the electronic structure from a semimetallic one to an n-type semiconductor-like one, with the emergence of high- T_c superconductivity. Here, the semimetallic and n-type semiconductor-like band structures are denoted as the N phase and the S phase, respectively, referring to the definition used in previous ARPES studies^{25,26}.

In this section, we analyse the d dependence of ρ_{2D} and the Seebeck coefficient α in FeSe. The continuous decrease of α of Sample A in Supplementary Fig. 3a, which is the same as Fig. 2b, suggests that the area of the S phase gradually expands with decreasing d , as schematically shown in Supplementary Figs. 3e-3g. Indeed, evidence of the phase separation was observed in the temperature dependence of ρ_{2D} since the N phase and the S phase have different ground states^{25,26,28–30}. Supplementary Figure 3b shows ρ_{2D} at $V_G = 5$ V normalized to the values at 200 K. When $d \sim 18$ nm, Sample A was insulating, with an upturn of ρ_{2D} at low temperatures. The superconducting transition occurred at ~ 40 K when d was ~ 15.8 nm, whereas the zero resistance was not observed. This clearly indicates the phase separation of the superconducting S and insulating N regions at intermediate thickness, as schematically shown in Supplementary Fig. 3f. Zero resistance was observed when d was reduced to ~ 11.5 nm, suggesting that the S regions were connected from one edge of the sample to the other.

The area ratio between the S phase and the N phase on the top surface of

Sample A is roughly estimated as follows. As shown in Supplementary Fig. 3b, the thinner FeSe film showed more metallic transport. This means that the S phase, which is dominant in thin regions, has larger metallic conductivity than that of the N phase. Therefore, $1 - \rho_{2D}^{100K} / \rho_{2D}^{200K}$, which represents the degree of the metallic transport, indicates the proportion of the S and the N phases on the top surface of FeSe. Supplementary Figure 1c shows the variation of $1 - \rho_{2D}^{100K} / \rho_{2D}^{200K}$, which should increase with increasing area of the S phase. Indeed, the value of $1 - \rho_{2D}^{100K} / \rho_{2D}^{200K}$ increased from ~ 0.1 at $d \sim 18$ nm to ~ 0.6 below ~ 8.5 nm. The saturation of $1 - \rho_{2D}^{100K} / \rho_{2D}^{200K}$ suggests that the surface of Sample A was mostly covered by the S phase below ~ 8.5 nm, as schematically shown in Supplementary Fig. 3g. Supplementary Figure 3d shows the d dependence of ρ_{2D}^{200K} . The value of ρ_{2D}^{200K} increased with decreasing d from ~ 18 nm, decreased below ~ 11 nm, and became roughly constant below ~ 8 nm. The reason that the R_s decreased below ~ 11 nm is that the area of the S phase, which is more conducting than the N phase, expanded with decreasing d . The fact that ρ_{2D}^{200K} stayed constant below ~ 8 nm suggests that most of the FeSe surface was the S phase below ~ 8 nm, which is consistent with the analysis of $1 - \rho_{2D}^{100K} / \rho_{2D}^{200K}$ in Supplementary Fig. 3c. The phase separation possibly originated from the thickness fluctuation of the initial condition¹⁶ and of the accumulation layer width for gate-induced carriers.

In order to confirm the discussion above, we performed a simple simulation on the d dependence of α . We consider that the overall volume of Sample A is the N phase and the S phase in the thick and thin limits, respectively. Therefore, α of the pure S phase, α_S , is about $-454 \mu\text{V K}^{-1}$, and that of the pure N phase, α_N , is about $+3.8 \mu\text{V K}^{-1}$.

Here, the simulation is performed based on a model where i) the phase separation between the N and the S occurs only on the gated top layer with the thickness d_{EDL} and ii) the carriers flow in a parallel circuit consisting of the gated top layer (with the two dimensional resistivity $\rho_{2D,T}$ and the Seebeck coefficient α_T) and the ungated bottom layer (with $\rho_{2D,B}$ and α_B). As for i), the total area of S phase, A_S , monotonically increased with decreasing d from ~ 18 nm to ~ 8.5 nm, as shown in Supplementary Fig. 3c. However, there is no information on the ratio of A_S and the total surface area A for $8.5 \text{ nm} < d < 18 \text{ nm}$. Therefore, we simply assumed A_S/A as follows

$$\frac{A_S}{A} = -\frac{d - 8.5}{18 - 8.5} + 1. \quad (\text{S2})$$

By further assuming that the S and N phases are connected only in series, the total Seebeck coefficient α is described as,

$$\alpha = \frac{\rho_{2D,T}\alpha_B + \rho_{2D,B}\alpha_T}{\rho_{2D,B} + \rho_{2D,T}}. \quad (\text{S3})$$

Here, α_T , $\rho_{2D,B}$, and $\rho_{2D,T}$ are described using A_S as follows:

$$\begin{aligned} \alpha_T &= \alpha_S \frac{A_S}{A} + \alpha_N \frac{A_N}{A} \\ &= \alpha_N + (\alpha_S - \alpha_N) \frac{A_S}{A} \end{aligned} \quad (\text{S4})$$

$$\rho_{2D,B} = \frac{\rho_B}{d - d_{EDL}} \quad (\text{S5})$$

$$\begin{aligned} \rho_{2D,T} &= \rho_{2D,S} \frac{A_S}{A} + \rho_{2D,N} \frac{A_N}{A} \\ &= \rho_{2D,N} + (\rho_{2D,S} - \rho_{2D,N}) \frac{A_S}{A} \\ &= \frac{\rho_N}{d_{EDL}} + \left(\frac{\rho_S}{d_{EDL}} - \frac{\rho_N}{d_{EDL}} \right) \frac{A_S}{A} \end{aligned} \quad (\text{S6})$$

Here, ρ_B , ρ_S , and ρ_N ($\sim \rho_B$) are the three dimensional resistivity of the ungated bottom

layer, the S phase, and the N phase, respectively. By substituting equations S2, S4, S5, and S6 into S3 with $\alpha_B = \alpha_N$, the d dependence of α is given. We take $d_{EDL} \sim 1.5$ nm in the simulation, since typical values of d_{EDL} in electric double layer transistors is 1~2 nm³¹⁻³⁴.

We found that the variation of α of Sample A was well-reproduced by the simulation, as shown by the solid curve in Supplementary Fig. 3a, which assures the validity of our model. The value of α shows a steep decrease with decreasing d from 18 nm to ~ 8.5 nm due to the expansion of the surface S region on the surface. Below $d \sim 8.5$ nm, α changes weakly but still keeps decreasing because the observed thermoelectric response includes contributions from the gated top surface and the ungated underneath layer. It should be noted that the assumptions on the surface S ratio for $8.5 \text{ nm} < d < 18 \text{ nm}$ (Eq. S2) and the connection of the S and N phases on the top surface (Eq. S4) do not affect the result at $d < 8.5$ nm. If more realistic expressions for Eqs. S2 and S4 were used, we would obtain a minor change in the d dependence of α only for $8.5 \text{ nm} < d < 18 \text{ nm}$.

Supplementary Note 3: Transfer characteristics of FeSe-based ion-gated transistor

The V_G dependence of ρ_{2D} for a FeSe thin film on a SrTiO₃ substrate, Sample D, detected interesting behaviour when a positive V_G was applied. Supplementary Figures 4a and 4b show ρ_{2D} normalized with the resistance ρ_{2D}^{0V} at $V_G = 0$ V. The gate bias V_G was swept at 220 K to electrostatically control the charge carrier density^{35,36} on the top surface of the FeSe film. Unexpectedly, the V_G dependence of ρ_{2D}/ρ_{2D}^{0V} considerably

changed just by decreasing d . As shown in Supplementary Fig. 4a, ρ_{2D}/ρ_{2D}^{0V} increased with increasing V_G when d was larger than ~ 10 nm, which agrees with the previous studies on FeSe thin films³⁶ and single crystals³⁵ with d of ~ 10 nm. This suggests that the dominant transport carriers of the FeSe film at 220 K are holes. When d was smaller than ~ 9 nm, on the other hand, the positive V_G reduced R_s (Supplementary Fig. 4b), suggesting that the dominant carriers in the thinner condition are electrons. According to the recent ARPES studies on FeSe, heavy electron doping into thin layers of FeSe by high-temperature annealing^{24–26} or K coating²⁷ does not simply induce a shift of E_F ; the electronic band structure is dramatically modulated, from a semimetallic one schematically shown in Supplementary Fig. 4c to an n-type semiconductor-like one in Supplementary Fig. 4d, with the emergence of high- T_c superconductivity. The V_G dependences of ρ_{2D}/ρ_{2D}^{0V} in Supplementary Fig. 4b and of α in Fig. 2b strongly support the idea that such an evolution of the band structure is caused by the field effect carrier doping as well (see Supplementary Note 2).

Supplementary Note 4: Calculation of Seebeck coefficient for N and S phases

In order to interpret the large negative Seebeck coefficient observed in the experiment, here we calculate the Seebeck coefficient based on the band structures calculated for bulk FeSe (the N phase) and monolayer FeSe (the S phase). The Seebeck coefficient is given by

$$\alpha = \frac{1}{-eT} \frac{K_1}{K_0} \quad (\text{S9})$$

where we define

$$K_n = \int d\epsilon L(\epsilon) (\epsilon - \mu)^n \left(-\frac{\partial f}{\partial \epsilon} \right) \quad (\text{S10})$$

and

$$L(\epsilon) = \sum_{\mathbf{k}, l} v_{kx}^l{}^2 \left(\frac{\eta}{(\epsilon + \mu - \epsilon_{\mathbf{k}}^l)^2 + \eta^2} \right)^2. \quad (\text{S11})$$

Here, μ denotes the chemical potential, f denotes the Fermi distribution function, $\epsilon_{\mathbf{k}}^l$ is the energy level for orbital l , $v_{kx}^l = \frac{\partial}{\partial k_x} \epsilon_{\mathbf{k}}^l$ is the band velocity, and η is the energy-broadening factor. For bulk FeSe, we employ a modified band dispersion (Supplementary Fig. 5a) so as to match the ARPES results³⁷. For monolayer FeSe, we first assume that the system is 2D. Then, in order to estimate the possible maximum Seebeck coefficient realized in the S phase, we consider the situation that only electron pockets contribute to the Seebeck coefficient, i.e., we use the band structure plotted in Supplementary Figs. 5b and 5c. This band structure is obtained by multiplying the renormalization factor 0.82 to the hopping integrals for d_{zx} and d_{yz} orbitals between nearest neighbour sites and eliminating the hole bands.

Supplementary Figure 1 Evaluation of contribution of SrTiO₃ substrates. **a**, Conventional configuration of electric double layer transistor based on SrTiO₃. The blue area of SrTiO₃ just beneath the Ti/Au contact electrodes was exposed to the Ar-ion milling to form conducting SrTiO_{3- δ} . **b**, Electric double layer transistor structure with

FeSe contact electrode. There is no $\text{SrTiO}_{3-\delta}$ layer below FeSe. **c**, Transfer characteristics of ion-gated SrTiO_3 . The FeSe contact electrode completely suppressed the drain-source current I_D . **d**, Estimation of Seebeck effect for virtual parallel conduction. $\rho_{2D,\text{STO}}$ and α_{STO} are the experimental values of the two dimensional resistivity and the Seebeck coefficient of an ion-gated SrTiO_3 at 200 K⁶. α_{STO} is the total Seebeck coefficient of FeSe and SrTiO_3 , which was estimated by Eq. S1. **e**, Temperature dependence of Seebeck coefficient in FeSe thin film and chemically-doped SrTiO_3 . The data of FeSe is the same as that of Fig. 4a in the main text. The data of La-doped SrTiO_3 were taken from literature^{20,21}.

Supplementary Figure 2 Seebeck effect of FeSe/KTaO₃. **a**, Thermoelectric voltage ΔV under temperature difference ΔT in an FeSe thin film on a KTaO₃ substrate, Sample C. The thermoelectric measurements were performed at 200 K with $V_G = 4.3$ V. The value of ΔV dramatically increased with decreasing thickness d from ~ 14.5 nm. **b**, Thickness d dependence of the Seebeck coefficient α . The values of α were estimated from the slope of the $\Delta V - \Delta T$ plot in **a** as $\alpha = -\Delta V / \Delta T$. With reducing the thickness, the absolute value of α was enhanced. The data for Sample A, which is the same with that in Fig. 2b, are also plotted in Supplementary Fig. 2b.

Supplementary Figure 3 Evolution of electronic structure. **a**, Thickness d dependence of Seebeck coefficient α in FeSe thin film, Sample A. The data is the same as those used in Fig. 2b. The thermoelectric measurements were performed at 200 K with $V_G = 5$ V. The absolute value of α showed a dramatic increase with decreasing d . The solid line is the result of the simulation. **b**, Temperature dependence of two dimensional resistivity ρ_{2D} . We normalized ρ_{2D} to the value at 200 K, $\rho_{2D}^{200\text{ K}}$. **c**, Development of S phase on FeSe top surface. The value of $1 - \rho_{2D}^{100\text{ K}}/\rho_{2D}^{200\text{ K}}$ increased with decreasing d and saturated at $d \sim 8$ nm, reflecting the evolution of the area of the S phase. **d**, Variation of $\rho_{2D}^{200\text{ K}}$ as a function of d . **e, f, g**, Schematic illustrations of phase separation of FeSe surface at $V_G = 5$ V.

Supplementary Figure 4 Surface carrier doping by electric field effect. a,b, Gate voltage V_G dependence of the two dimensional resistivity ρ_{2D} for Sample D. The thickness of the film was tuned by the electrochemical etching technique¹⁶. Here, ρ_{2D} was normalized to the resistivity ρ_{2D}^{0V} at $V_G = 0$ V. The bias V_G was swept at 220 K in order to electrostatically increase the electron carrier density of the top surface of Sample D^{35,36}. **c,d,** Schematic band structures of FeSe. The field effect carrier doping induces the reconstruction of the band structure from **c** to **d**, especially in the region thinner than ~ 9 nm.

Supplementary Figure 5 Calculated band structures of FeSe. We calculated the band structures of **a** bulk systems and **b** thin films. The enlarged view of area outlined by the dashed lines in **b** is shown in **c**.

Supplementary Table 1 Reference list of thermoelectric materials referred in Fig.

3c. The values of the power factor for Na_xCoO_2 ³⁸, $\text{Bi}_{1-x}\text{Sb}_x$ ³⁹, Bi_2Te_3 ⁴⁰, $(\text{Sr},\text{La})\text{TiO}_3$ ⁴¹, CsBi_4Te_6 ⁴², Ta_4SiTe_4 ⁴³, FeSb_2 ⁴⁴, which are plotted in Fig. 3c, were cited from the literatures in this table.

Material	Reference
Na_xCoO_2	M. Lee et al. , Nature Mater. 5, 537 (2006).
$\text{Bi}_{1-x}\text{Sb}_x$	B. Lenoir et al. , J. Phys. Chem. Solids 59, 129 (1998).
Bi_2Te_3	V. A. Kulbachinskii et al. , J. Solid State Chem. 193, 47 (2012).
$(\text{Sr},\text{La})\text{TiO}_3$	J. Fukuyado et al. , Phys. Rev. B 85, 075112 (2012).
CsBi_4Te_6	D. Chung et al. , Science 287, 1024 (2000).
Ta_4SiTe_4	T. Inohara et al. , Appl. Phys. Lett. 110, 183901(2017).
FeSb_2	H. Takahashi et al. , Nature Commun. 7, 12732 (2016).

Supplementary References

1. Ueno, K. *et al.* Electric-field-induced superconductivity in an insulator. *Nature Mater.* **7**, 855–858 (2008).
2. Lee, Y. *et al.* Phase Diagram of Electrostatically Doped SrTiO₃. *Phys. Rev. Lett.* **106**, 136809 (2011).
3. Li, M., Graf, T., Schladt, T. D., Jiang, X. & Parkin, S. S. P. Role of Percolation in the Conductance of Electrolyte-Gated SrTiO₃. *Phys. Rev. Lett.* **109**, 196803 (2012).
4. Li, M. *et al.* Suppression of ionic liquid gate-induced metallization of SrTiO₃(001) by oxygen. *Nano Lett.* **13**, 4675–4678 (2013).
5. Gallagher, P., Lee, M., Williams, J. R. & Goldhaber-Gordon, D. Gate-tunable superconducting weak link and quantum point contact spectroscopy on a strontium titanate surface. *Nature Phys.* **10**, 748–752 (2014).
6. Shimizu, S., Ono, S., Hatano, T., Iwasa, Y. & Tokura, Y. Enhanced cryogenic thermopower in SrTiO₃ by ionic gating. *Phys. Rev. B* **92**, 165304 (2015).
7. Stanwyck, S. W., Gallagher, P., Williams, J. R. & Goldhaber-gordon, D. Universal Conductance Fluctuations in Electrolyte-Gated SrTiO₃ Nanostructures. *Appl. Phys. Lett.* **103**, 213504 (2013).
8. Reagor, D. W. & Butko, V. Y. Highly conductive nanolayers on strontium titanate produced by preferential ion-beam etching. *Nature Mater.* **4**, 593–596 (2005).
9. Atesci, H. *et al.* On the formation of a conducting surface channel by ionic liquid gating of an insulator. *arXiv* :1709.01178v1 (2017).

10. Wu, C. T. *et al.* Heterojunction of Fe(Se_{1-x}Te_x) superconductor on Nb-doped SrTiO₃. *Appl. Phys. Lett.* **96**, 122506 (2010).
11. Zhang, W. *et al.* Interface charge doping effects on superconductivity of single-unit-cell FeSe films on SrTiO₃ substrates. *Phys. Rev. B* **89**, 060506(R) (2014).
12. Zhang, H. *et al.* Origin of charge transfer and enhanced electron-phonon coupling in single unit-cell FeSe films on SrTiO₃. *Nature Commun.* **8**, 214 (2017).
13. Wang, Q.-Y. *et al.* Interface-induced high-temperature superconductivity in single unit-cell FeSe films on SrTiO₃. *Chinese Phys. Lett.* **29**, 37402 (2012).
14. Sun, Y. *et al.* High temperature superconducting FeSe films on SrTiO₃ substrates. *Sci. Rep.* **4**, 6040 (2014).
15. Wang, Q. *et al.* Thickness dependence of superconductivity and superconductor-insulator transition in ultrathin FeSe films on SrTiO₃ (001) substrate. *2D Mater.* **2**, 44012 (2015).
16. Shiogai, J., Ito, Y., Mitsuhashi, T., Nojima, T. & Tsukazaki, A. Electric-field-induced superconductivity in electrochemically etched ultrathin FeSe films on SrTiO₃ and MgO. *Nature Phys.* **12**, 42–46 (2016).
17. Arsenijević, S. *et al.* Pressure effects on the transport coefficients of Ba(Fe_{1-x}Co_x)₂As₂. *Phys. Rev. B* **84**, 75148 (2011).
18. Zhu, Z. W. *et al.* Nernst effect of a new iron-based superconductor LaO_{1-x}F_xFeAs. *New J. Phys.* **10**, 63021 (2008).
19. Kondrat, A., Behr, G., Büchner, B. & Hess, C. Unusual Nernst effect and spin density wave precursors in superconducting LaFeAsO_{1-x}F_x. *Phys. Rev. B* **83**,

- 92507 (2011).
20. Ohta, S., Nomura, T., Ohta, H. & Koumoto, K. High-temperature carrier transport and thermoelectric properties of heavily La- or Nb-doped SrTiO₃ single crystals. *J. Appl. Phys.* **97**, 34106 (2005).
 21. Cain, T. A., Kajdos, A. P. & Stemmer, S. La-doped SrTiO₃ films with large cryogenic thermoelectric power factors. *Appl. Phys. Lett.* **102**, 182101 (2013).
 22. Wang, G., Endicott, L., Chi, H., Lošt'ák, P. & Uher, C. Tuning the Temperature Domain of Phonon Drag in Thin Films by the Choice of Substrate. *Phys. Rev. Lett.* **111**, 46803 (2013).
 23. Suemune, Y. Thermal Conductivity of BaTiO₃ and SrTiO₃ from 4.5° to 300°K. *J. Phys. Soc. Jpn.* **20**, 174–175 (1965).
 24. Liu, D. *et al.* Electronic origin of high-temperature superconductivity in single-layer FeSe superconductor. *Nature Commun.* **3**, 931 (2012).
 25. He, S. *et al.* Phase diagram and electronic indication of high-temperature superconductivity at 65 K in single-layer FeSe films. *Nature Mater.* **12**, 605–610 (2013).
 26. He, J. *et al.* Electronic Evidence of an Insulator-Superconductor Transition in Single-Layer FeSe/SrTiO₃ Films. *Proc. Natl. Acad. Sci. U. S. A.* **111**, 18501–18506 (2014).
 27. Miyata, Y., Nakayama, K., Sugawara, K., Sato, T. & Takahashi, T. High-temperature superconductivity in potassium-coated multilayer FeSe thin films. *Nature Mater.* **14**, 775–779 (2015).
 28. Ye, Z. R. *et al.* Simultaneous emergence of superconductivity, inter-pocket

- scattering and nematic fluctuation in potassium-coated FeSe superconductor. *arXiv* :1512.02526 (2015).
29. Wen, C. H. P. *et al.* Anomalous correlation effects and unique phase diagram of electron-doped FeSe revealed by photoemission spectroscopy. *Nature Commun.* **7**, 10840 (2016).
 30. Shi, X. *et al.* Enhanced superconductivity accompanying a Lifshitz transition in electron-doped FeSe monolayer. *arXiv* :1606.01470 (2016).
 31. Uesugi, E., Goto, H., Eguchi, R., Fujiwara, A. & Kubozono, Y. Electric double-layer capacitance between an ionic liquid and few-layer graphene. *Sci. Rep.* **3**, 1595 (2013).
 32. Chu, L. *et al.* Charge transport in ion-gated mono-, bi-, and trilayer MoS₂ field effect transistors. *Sci. Rep.* **4**, 7293 (2014).
 33. Saito, Y., Kasahara, Y., Ye, J., Iwasa, Y. & Nojima, T. Metallic ground state in an ion-gated two-dimensional superconductor. *Science* **350**, 409–413 (2015).
 34. Saito, Y. *et al.* Superconductivity protected by spin–valley locking in ion-gated MoS₂. *Nature Phys.* **12**, 144–149 (2016).
 35. Lei, B. *et al.* Evolution of High-Temperature Superconductivity from a Low T_c Phase Tuned by Carrier Concentration in FeSe Thin Flakes. *Phys. Rev. Lett.* **116**, 77002 (2016).
 36. Hanzawa, K., Sato, H., Hiramatsu, H., Kamiya, T. & Hosono, H. Electric field-induced superconducting transition of insulating FeSe thin film at 35 K. *Proc. Natl. Acad. Sci. U. S. A.* **113**, 3986–3990 (2016).
 37. Suzuki, Y. *et al.* Momentum-dependent sign inversion of orbital order in

- superconducting FeSe. *Phys. Rev. B* **92**, 205117 (2015).
38. Lee, M. *et al.* Large enhancement of the thermopower in Na_xCoO_2 at high Na doping. *Nature Mater.* **5**, 537–540 (2006).
 39. Lenoir, B., Dauscher, A., Cassart, M., Ravichch, Y. I. & Scherrer, H. Effect of antimony content on the thermoelectric figure of merit of $\text{Bi}_{1-x}\text{Sb}_x$ alloys. *J. Phys. Chem. Solids* **59**, 129–134 (1998).
 40. Kulbachinskii, V. A., Kytin, V. G., Kudryashov, A. A. & Tarasov, P. M. Thermoelectric properties of Bi_2Te_3 , Sb_2Te_3 and Bi_2Se_3 single crystals with magnetic impurities. *J. Solid State Chem.* **193**, 47–52 (2012).
 41. Fukuyado, J., Narikiyo, K., Akaki, M., Kuwahara, H. & Okuda, T. Thermoelectric properties of the electron-doped perovskites $\text{Sr}_{1-x}\text{Ca}_x\text{Ti}_{1-y}\text{Nb}_y\text{O}_3$. *Phys. Rev. B* **85**, 75112 (2012).
 42. Chung, D. *et al.* CsBi_4Te_6 : A high-performance thermoelectric material for low-temperature applications. *Science* **287**, 1024–1027 (2000).
 43. Inohara, T., Okamoto, Y., Yamakawa, Y., Yamakage, A. & Takenaka, K. Large Thermoelectric Power Factor at Low Temperatures in One-Dimensional Telluride Ta_4SiTe_4 . *Appl. Phys. Lett.* **110**, 183901 (2017).
 44. Takahashi, H. *et al.* Colossal Seebeck effect enhanced by quasi-ballistic phonons dragging massive electrons in FeSb_2 . *Nature Commun.* **7**, 12732 (2016).

Reviewers' comments:

Reviewer #1 (Remarks to the Author):

The comments have been considered carefully. However, there are still some points need to be clarified.

(1) For question 1, the authors attribute the enhancement in Seebeck coefficient to the open of band at M point (Figure R2), because of the electron doping caused by gate bias VG. Is there any direct evidence? In addition, we believe the Seebeck coefficient increases with the open of band gap (semimetallic to semiconductor), why the electrical conductivity also sharply increases with the open of band gap (Figure 2c)? It is amazing and unusual.

For question 4, the authors pointed out the resistivity of the $d=18$ nm samples with $V_G=0$, however, what is the resistivity values for $d=1-5$ nm measured without VG (as shown in comments 4). The authors may not notice this question

Reviewer #2 (Remarks to the Author):

I have read the authors' response and the revised manuscript. I think the revised manuscript is now ready as the publication of Nature Communications.

Response to Reviewers' Comments

First of all, we thank Reviewers for the second round reviewing of our manuscript. We have revised the manuscript according to the comments given by Reviewer#1 and would like to reply to all the comments in the followings. For the reviewers' convenience, we highlighted the revised part in yellow in the main text.

* Reply to comments of Reviewer#1 *

[Comment 0]

The comments have been considered carefully. However, there are still some points need to be clarified.

[Reply 0]

We would like to thank Reviewer#1 for the careful reading of our manuscript for the second round reviewing. We would like to try our best to properly reply all the comments one by one.

[Comment 1]

For question 1, the authors attribute the enhancement in Seebeck coefficient to the open of band at M point (Figure R2), because of the electron doping caused by gate bias VG. Is there any direct evidence?

[Reply 1]

As commented by Reviewer#1, we discussed the origin of the enhancement of the Seebeck effect in Discussion section and attributed it to the band gap opening at M point. This discussion is based on our previous report on the Hall effect measurements of the ion-gated FeSe thin films [Shiogai et al., Phys Rev. B 95, 115101 (2017).] and the other groups' published papers on ARPES measurements of FeSe/SrTiO₃ systems.

The superconducting transition temperature $T_c = 8$ K in bulk FeSe is enhanced to 40 K or much higher (upto ~ 100 K) by not only the ionic liquid gating but also fabricating monolayer FeSe on SrTiO₃ substrate [Wang *et al.*, Chinese Phys. Lett. 29, 037402 (2012).; He *et al.*, Nature Mater. 12, 605 (2013).] and coating the surface of FeSe thin films with K ions [Miyata *et al.*, Nature Mater. 14, 775 (2015).; Wen *et al.*, Nature Commun. 7, 10840 (2016).]. Importantly, all the three different structures commonly dope the similar amount of electrons, $\sim 10^{14}$ cm⁻², into the monolayer or nanometer thick regions of FeSe, resulting in the abrupt enhancement of T_c . This means that the same electronic structure is expected in the ion-gated FeSe thin films, the monolayer FeSe/SrTiO₃, and the K coated FeSe thinfilms.

We reported the Hall effect measurements on the ion-gated FeSe thin films in the previous study. Actually, we observed that the Hall coefficient changes from positive to negative values by ionic gating [Shiogai *et al.*, Phys Rev. B 95, 115101 (2017).], which suggests that the dominant carrier changes from holes to electrons due to the electron accumulation in the conduction band at around M point. On the other hand, the ARPES measurements on the monolayer FeSe/SrTiO₃ and the K coated FeSe thin films revealed that the high- T_c phase is induced by electron doping and that the band gap is opened at M point due to the band reconstruction triggered by the electron doping. Although the mechanism of electron doping is different, the resultant band structure should be similar among our ion-gated FeSe thin films, the monolayer FeSe, and the K coated FeSe thin films. This is because similar T_c enhancement is observed by doping similar amount of electrons.

We would like to mention the basis that the band gap opening is expected in the ion-gated FeSe thin films in the Discussion session as follows: “This behavior is consistent with the band structure evolution derived from the angle-resolved photoemission spectroscopy (ARPES)^{12,35–37} in monolayer FeSe on SrTiO₃ and in K coated FeSe thin films, which clarified that the hole pocket at the Γ -point disappears and a gap of ~ 60 meV is opened at the M-point³⁵ due to the thinning and concomitant electron doping. The present ion-gated FeSe thin films should have a similar band structure because the electron density accumulated by the ionic gating, $\sim 10^{14}$ cm⁻², is comparable to that of the charge transfer from SrTiO₃ substrate and

of surface K coating.”

[Comment 2]

In addition, on page 8 line 131, the electrical conductivity $\sigma=1/(R \times d)$ is incorrect. In addition, we believe the Seebeck coefficient increases with the open of band gap (semimetallic to semiconductor), why the electrical conductivity also sharply increases with the open of band gap (Figure 2c)? It is amazing and unusual.

[Reply 2]

We do not use the electrical conductivity σ in the revised manuscript in order to avoid ambiguity regarding to the definition. Instead, we use the two dimensional sheet resistance ρ_{2D} , which is normalized by the length and the width of the thin film, and the three dimensional resistivity ρ . The relationship among σ [$1/(\Omega\text{cm})$], ρ [Ωcm], and ρ_{2D} [Ω] is $\sigma = 1/\rho = 1/(\rho_{2D} \times d)$, where d is the film thickness.

As for the reduction of ρ_{2D} in Figure 2c, the application of V_G induces not only the film thinning but also electron doping on the FeSe film. The former results in the band gap opening at the M point as discussed in [Replay 1], and the latter causes the reduction of ρ_{2D} in Figure 2c. It is these characteristics that make FeSe ultra thin films a very unique thermoelectric semiconductor. Such an evolution in band structures has never been observed in conventional semiconductors; this can be an emergence of the novel functionality that reflects two dimensional natures.

We have added the sentence “Here, it is noted that the application of V_G not only induces the electrochemical etching of the thin films but also accumulates the electron carriers on the top surface.” in the first paragraph of the Results section.

[Comment 3]

For question 4, the authors pointed out the resistivity of the $d=18$ nm samples with $V_G=0$, however, what is the resistivity values for $d=1-5$ nm measured without V_G (as shown in comments 4). The authors may not notice this question

[Reply 3]

We would like to thank the reviewer for commenting on this issue because the clear description of the thickness dependence experiments is beneficial for readers. The Comment (question 4) given by Reviewer#1 in the first round reviewing was **“Given no electrical resistivity data was provided for samples with thickness between 1-5 nm and 10-18 nm measured without gate bias, thus ‘the sheet resistance at 200 K, $R_{s200 K}$, first increased with decreasing d for 120 both $V_G = 0 V$ and $5 V$, as expected’ is unconvincing.”**. As mentioned by Reviewer#1, no data were taken for $d = 1 - 5$ nm at $V_G = 0$ V. Therefore, we removed the phrase “as expected” from the sentence pointed out by Reviewer#1 in the second paragraph of the Results section so that we explain only the experimental facts. Here, we would like to explain the thickness dependence measurements in detail.

In Figure 2b and 2c, we showed the thickness d dependence of the Seebeck coefficient and the sheet resistance, respectively. The data at $V_G = 0$ V were taken only for $d \sim 18$ nm, 11 nm, 8 nm, and 6 nm.

The reason that we measured the Seebeck coefficient and the sheet resistance at $V_G = 0$ V for the relatively thick films is the followings. First, we aimed at ruling out the possibility that the large Seebeck effect originates from oxygen deficiencies in the SrTiO₃ substrate that might be formed in the thin film fabrication process. Second, we intended to exclude another possibility that the electron charge transfer from SrTiO₃ to FeSe is the origin of the large Seebeck effect. Finally, we wanted to show that both the high- T_c superconductivity and the large Seebeck effects are concomitantly induced by V_G . For those purposes, we measured the Seebeck coefficient and the resistance at $V_G = 0$ V for the four different thicknesses. We found that when V_G is switched off, the resistance becomes much higher and the Seebeck coefficient becomes extremely small. This indicates that neither the oxygen vacancies nor charge transfer from SrTiO₃ to FeSe are playing predominantly roles in both superconductivity and the Seebeck effect enhancement, but only the V_G application is the key factor.

We sincerely hope that this reply has reasonably answered to the comment given by Reviewer#1 on the thickness dependence measurements.

*** Reply to comments of Reviewer#2 ***

[Comment 0]

I have read the authors' response and the revised manuscript. I think the revised manuscript is now ready as the publication of Nature Communications.

[Reply 0]

We would like to thank Reviewer#2 for the careful reading of our manuscript again and for suggesting publication of our paper in Nature Communications.

REVIEWERS' COMMENTS:

Reviewer #1 (Remarks to the Author):

the manu can be accepted

Response to Reviewers' Comments

Once again, we would like to thank both Reviewer#1 and Reviewer#2, who carefully reviewed our manuscript. We are sure that the manuscript has been improved and much clearer to readers. In addition, we are very glad that our previous replies and revisions cleared up the remaining concerns of Reviewer#1. We have formatted the manuscript to comply with the editorial requirements, hoping that the current version is acceptable for publication in Nature Communications.

*** Reply to comments of Reviewer#1 ***

[Comment]

the manu can be accepted

[Reply]

We would like to thank Reviewer#1 for the careful reading of our manuscript again and for suggesting acceptance of our paper in Nature Communications.